# Efficient cross-trait penalized regression increases prediction accuracy in large cohorts using secondary phenotypes

Wonil Chung [1,2], Jun Chen[3], Constance Turman[1,2], Sara Lindstrom[4], Zhaozhong Zhu [1,2,5], Po-Ru Loh[1,2,6], Peter Kraft[1,2,7] & Liming Liang [1,2,7]

We introduce cross-trait penalized regression (CTPR), a powerful and practical approach for multi-trait polygenic risk prediction in large cohorts. Specifically, we propose a novel cross-trait penalty function with the Lasso and the minimax concave penalty (MCP) to incorporate the shared genetic effects across multiple traits for large-sample GWAS data. Our approach extracts information from the secondary traits that is beneficial for predicting the primary trait based on individual-level genotypes and/or summary statistics. Our novel implementation of a parallel computing algorithm makes it feasible to apply our method to biobank-scale GWAS data. We illustrate our method using large-scale GWAS data (~1M SNPs) from the UK Biobank ($N = 456{,}837$). We show that our multi-trait method outperforms the recently proposed multi-trait analysis of GWAS (MTAG) for predictive performance. The prediction accuracy for height by the aid of BMI improves from $R^2 = 35.8\%$ (MTAG) to 42.5% (MCP + CTPR) or 42.8% (Lasso + CTPR) with UK Biobank data.

[1] Program in Genetic Epidemiology and Statistical Genetics, Harvard T.H. Chan School of Public Health, Boston, MA 02115, USA. [2] Department of Epidemiology, Harvard T.H. Chan School of Public Health, Boston, MA 02115, USA. [3] Division of Biomedical Statistics and Informatics and Center for Individualized Medicine, Mayo Clinic, Rochester, MN 55905, USA. [4] Department of Epidemiology, University of Washington, Seattle, WA 98195, USA. [5] Department of Environmental Health, Harvard T.H. Chan School of Public Health, Boston, MA 02115, USA. [6] Program in Medical and Population Genetics, Broad Institute of Harvard and MIT, Cambridge, MA 02142, USA. [7] Department of Biostatistics, Harvard T.H. Chan School of Public Health, Boston, MA 02115, USA. Correspondence and requests for materials should be addressed to L.L. (email: lliang@hsph.harvard.edu)

With the arrival of large-scale public biobanks harboring more than 500K samples, polygenic risk scores (PRS)-based methods[1–7] have been widely adopted for genetic risk prediction in practice due to computational feasibility and easy accessibility of the genome-wide association study (GWAS) summary statistics. PRS are usually constructed as a weighted sum of adjusted genetic effects through linkage disequilibrium (LD) information[8]. To boost predictive power of PRS, various approaches, including weighted multi-trait summary statistic best linear unbiased prediction method (wMT-SBLUP)[9] and multi-trait analysis of GWAS (MTAG)[10], incorporate information contained in related traits that share genetic architecture with a trait of interest. However, such approaches only utilize marginal single-nucleotide polymorphism (SNP) effects of multiple traits, not SNP effects conditional on other SNPs and thus may produce less accurate risk scores and limited prediction accuracy (PA) to some extent. There exists a necessity to develop multi-trait prediction methods using whole-genome individual-level genotypes.

The most commonly used whole-genome prediction methods for multiple traits are bivariate ridge regression method[11] and multi-trait genomic best linear unbiased prediction method (MTGBLUP)[12–15], which treat the genetic effects as random to obtain individual and SNP risk predictors using one or more genetically correlated traits. However, MTGBLUP requires the estimation of genetic relationship matrix (GRM), which becomes computationally prohibitive as the sample size increases. Furthermore, it implicitly assumes the infinitesimal genetic architecture, which indicates all variants are causal with relatively small effect size, whereas in reality complex traits or diseases are estimated to have roughly only a few thousand causal variants on the genome[16,17]. It deserves to construct polygenic risk predictors to accommodate a broad range of genetic architectures for large-sample datasets. Penalized regression methods such as the least absolute shrinkage and selection operator (Lasso)[18,19], the elastic net[20], the adaptive Lasso[21], the minimax concave penalty (MCP)[22,23], or statistical learning approaches[24] have previously been evaluated for genomic risk prediction[25,26]. These methods have the potential to be applied to GWAS data with large sample size via efficient penalized regression algorithm under the non-infinitesimal assumption. Furthermore, they can be extended to incorporate genetic effects for related traits via summary statistics by adding additional penalty functions[27,28] while the multivariate linear mixed model of MTGBLUP requires individual-level genotype data for all traits[12].

Here we develop a new statistical framework for cross-trait penalized regression (CTPR) for polygenic risk prediction of complex traits using individual-level data and/or summary statistics in large cohorts. In contrast to other methods based on multivariate modeling such as MTGBLUP and MTAG, our method attempts to optimize the PA only for the primary trait of interest. This strategy allows us to leverage information from other traits that is useful for the primary trait only, and thus additional traits with different degrees of genetic relatedness could be effectively utilized. Our approach takes advantage of genetic correlation among multiple traits to predict the primary trait of interest using penalized least-squares methods. Among many regularized methods, the Lasso and the MCP are evaluated for inducing a sparse solution. In order to incorporate the shared genetic effects across traits for improving PA, we propose a cross-trait penalty which is a smooth function of pairwise genetic effects. This penalty function utilizes not only individual-level genotypes but also summary statistics and thus the method can exploit many recent largest GWAS results. Because multiple traits are not required to be measured on the same individuals, any existing datasets of genetically correlated traits can be used to improve PA. Another important feature of our method is that our novel implementation of a parallel computing algorithm called Message Passing Interface (MPI) makes it feasible to apply our methods to large biobank-scale GWAS data. Based on the new computing algorithm, we apply our method to large-scale GWAS data (~1M) from UK Biobank[29–31] ($N = 456,837$) and NHS/HPFS/PHS cohort[32] ($N = 20,676$) and perform systematic simulation studies and real GWAS analyses. Using large-scale GWAS datasets, we show that our multi-trait methods significantly increase the PA, as illustrated for human height (HGT) by incorporating information from body mass index (BMI), hip circumference (HIP), waist circumference (WST), and waist–hip ratio (WHR) compared to other single-trait approaches. We further demonstrate that our prediction methods outperform other multi-trait approaches, including MTGBLUP and the recently proposed MTAG methods.

## Results

**Overview of methods**. The CTPR algorithm estimates all SNP effects for the primary trait of interest via a new multivariate penalized least-squares method utilizing information from individual-level data and/or summary statistics of genetically correlated secondary traits. We propose a new cross-trait quadratic penalty function with the Lasso or the MCP penalty to incorporate common genetic effects across multiple traits. This function induces the smoothness of the coefficients and can incorporate the prior knowledge on the similarity of a pair of traits at a given SNP via the adjacency coefficients. It effectively extracts information from the secondary traits that is beneficial for the primary trait but scales down information that is not. All coefficients are estimated in a computationally effective way using coordinate decent algorithm. We utilize $n$-fold cross-validation (e.g. $n = 5$) to select tuning parameters (i.e. $\lambda_1$ for Lasso or MCP and $\lambda_2$ for cross-trait penalty). The data for the primary trait are randomly partitioned into $n$ equal-sized subsets, where $n-1$ subsets are used as a training set and the remaining one as a validation set. We select the optimal values for $\lambda_1$ and $\lambda_2$ that minimize the averaged mean squared error (MSE) of the primary trait over $n$ folds. Because the closed form solution for coordinate descent algorithm over a single SNP coefficient exists, our method is computationally efficient and can be extended for parallel computing.

To make it feasible to apply our methods to biobank-based GWAS data, we further develop a novel distributed memory parallel computing algorithm utilizing MPI. First, large-scale GWAS data are divided into non-overlapping subgroups containing SNPs in low LD within each subgroup and each MPI core is assigned to one of subgroups with its own memory. Next, we define another group, called core-group, each of which contains several subgroups. All MPI cores in the same core-group run simultaneously at each estimation step keeping all cores in other core-groups waiting till finish. In this way, coefficients within a core-group are simultaneously updated and eventually all coefficients are updated consecutively in core-group to improve the computational efficiency as well as to avoid convergence problem. This algorithm enables multiple subgroups of SNP coefficients updated simultaneously or sequentially at each estimation step and therefore it provides the approximate (time-saving) or the exact (time-consuming) solution for coordinate descent optimization. We observed these approaches produce similar predictive power as long as the convergence is achieved. The overview of our CTPR method regarding MPI algorithm for biobank-based GWAS data is described in Fig. 1.

To assess the computational feasibility of CTPR for biobank-based GWAS data, we simulated data using $N = 437$K individuals

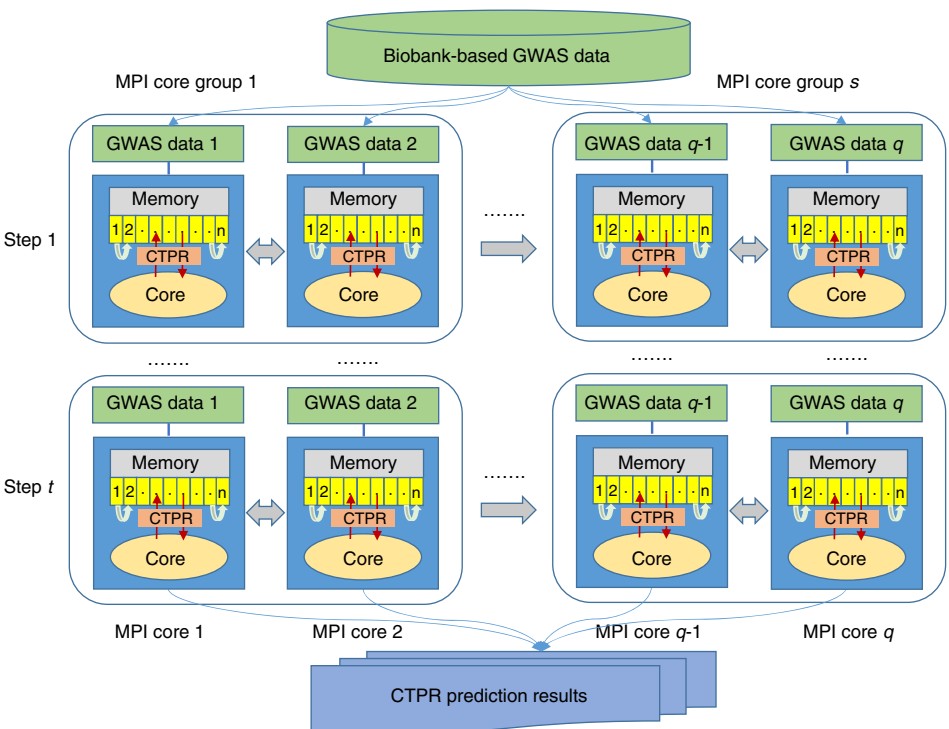

**Fig. 1** Overview of CTPR method regarding distributed memory MPI algorithm for biobank-based GWAS data. Biobank-based large-scale GWAS data are first divided into $q$ non-overlapping subgroups (GWAS data 1,...,$q$) containing SNPs in low LD and each MPI core is assigned to one of $q$ subgroups. Each subgroup of GWAS data runs on each MPI core of computing nodes with its own memory which only requires $1/q$ of whole memory size. We next propose to use another group, i.e. $s$ core-group ($s \leq q$), each of which contains several subgroups. All MPI cores in each core-group execute simultaneously at each estimation step (1,...,$t$) keeping all cores in other core-groups waiting till finish. In this way, coefficients within a core-group are concurrently updated and eventually all coefficients are updated consecutively in core-groups to improve the computational efficiency as well as to avoid convergence problem. This algorithm enables multiple subgroups of SNP coefficients updated simultaneously or sequentially at each estimation step and therefore it provides the computationally more efficient or exact coordinate descent optimization for polygenic risk prediction

and $P = 1$M SNPs from UK Biobank, which required ~1.7TB of memory with float data type (i.e. 437K*1M*4B = ~1.7TB). The CTPR ran on 40 cores (Intel Xeon CPU 2.1 GHz) with 48 GB of memory for each core, total of ~1.9 TB of memory, for up to 7 days to complete the analyses with 40 core-groups (exact solution). The running time of CTPR depends linearly not only on the sample size ($N$) and the number of SNPs ($P$) but also on the number of core-group ($q$), which represents $O(NPq)$. With 10 core-groups (approximate solution), the running time of CTPR dropped to ~1.75 days and it still generated almost the same predictive performance as exact solution due to good convergence. Even when sample size increases, the running time is able to remain similar because larger sample size increases likelihood of convergence and therefore less number of core-groups are needed.

**Predictive power of CTPR vs existing methods in simulations.**
To evaluate the predictive power of CTPR in a practical way, we first mimicked the real GWAS data (e.g. height, BMI) for simulation and then changed three important parameters (i.e. genetic correlation, number of traits, sample size) one at a time. These simulations were designed to provide a clear picture on how each parameter would affect the gain in power of CTPR and hence we can obtain practical insight on how to increase PA for the primary trait of interest through collecting appropriate secondary traits and more samples. We further compared our proposed methods with the existing methods such as MTAG and MTGBLUP under the same simulation settings.

We sampled 30,000 (30K), 200,000 (200K), or 436,837 (437K) individuals for a training set and 20,000 (20K) individuals for a validation set with 955,842 (1M) common SNPs from UK Biobank. We simulated up to four phenotypes with 6.8K–12.6K causal SNPs[17,33] explaining 45% of phenotypic variance for the primary trait and 25% for the secondary traits and varied genetic correlation among four traits ($\rho = 0.25, 0.5, 0.75$). To simulate the phenotypes, we used the classical linear model and generated genetic effects from a multivariate normal distribution with the desired SNP-heritability level and genetic correlation among multiple traits. The random error for each individual was independently sampled from a univariate normal distribution. PA was assessed throughout the paper using squares of correlation between true and predicted phenotype values, which is equivalent to prediction $R^2$.

We first considered the scenario with 30K training samples, two traits and ranged genetic correlation from 0.25 to 0.75 (Fig. 2a). For the secondary traits, our method utilized SNP effects summary statistics from SNPTEST[33] which fitted a single SNP linear regression with top 10 genotype PCs. The implementation of a novel cross-trait penalty function enables the CTPR to exploit fixed SNP effects of the secondary trait. As shown, increase in genetic correlation between two traits resulted in a gain in PA of multi-trait methods (Lasso + CTPR or MCP + CTPR) over single-trait methods (Lasso or MCP) in terms of prediction $R^2$. The Lasso + CTPR performed slightly better than the MCP + CTPR and the Lasso performed better than the MCP. We next examined if more secondary traits can improve PA of the primary trait (Fig. 2b). The genetic correlation between the primary trait

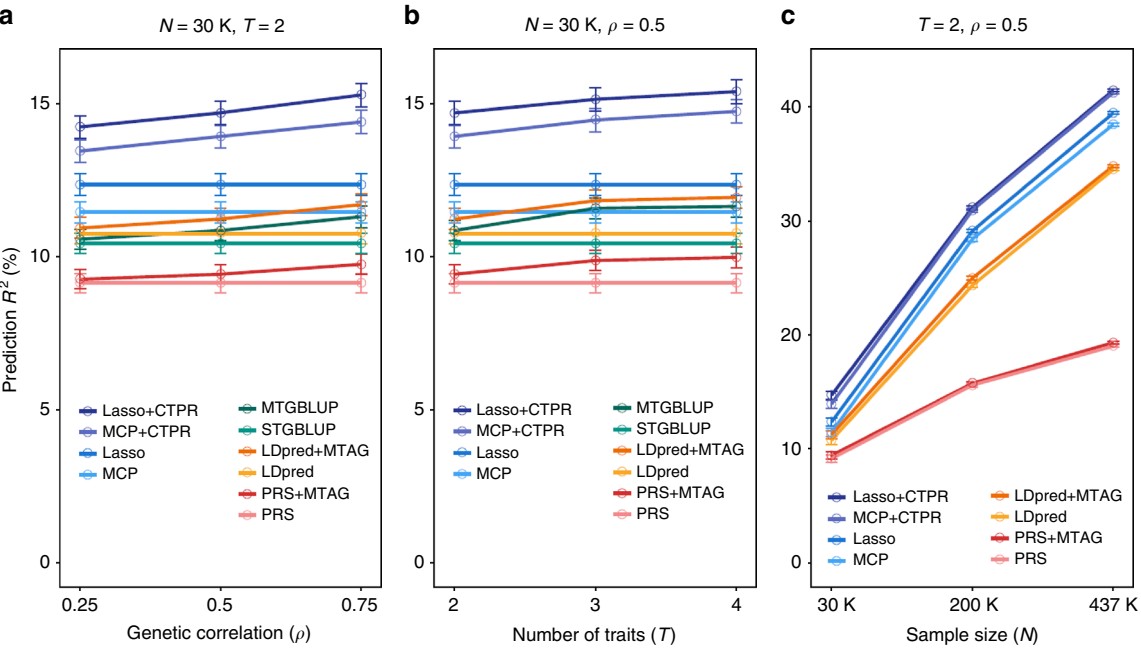

**Fig. 2** CTPR increases the predictive power over our single-trait methods (Lasso, MCP) and the existing methods such as LDpred, MTAG, STGBLUP, and MTGBLUP in simulation with various genetic correlation between two traits ($\rho$), number of traits ($T$), and sample size ($N$). Prediction $R^2$ were computed using LDpred, MTAG, STGBLUP, MTGBLUP, and CTPR with large-sample GWAS data from UK Biobank through varying genetic correlation ($\rho$), number of traits ($T$), and number of samples ($N$). We generated simulation data with 45% SNP-heritability, three different genetic correlations among traits ($\rho = 0.25, 0.5, 0.75$), three different sample sizes ($N = 30$K, 200K, 437K) and up to four traits. Clearly, our multi-trait methods (Lasso + CTPR, MCP + CTPR) outperformed our single-trait methods and the existing prediction methods under the same simulation settings. All error bars represent standard errors of prediction $R^2$. **a** Prediction $R^2$ were computed with $N = 30$K, $T = 2$ and varied $\rho$ from 0.25 to 0.75. **b** Prediction $R^2$ were computed with $N = 30$K, $\rho = 0.5$, and varied $T$ from 2 to 4. **c** Prediction $R^2$ were computed with $T = 2$, $\rho = 0.5$ and varied $N$ from 30K to 437K. It does not contain genetic prediction results of GBLUP methods because GBLUP methods are computationally infeasible with large sample size (e.g. $N = 200$K or 437K)

and each of the secondary traits was fixed to 0.5 and the genetic correlation among secondary traits was set to 0. The PA of multi-trait methods increased as more traits were included because SNP effects of the primary trait can be estimated more accurately from the correlated secondary traits. Next, we evaluated the effect of sample size on PA (Fig. 2c). As sample size increased from 30K to 437K, the PA of both single-trait and multi-trait methods dramatically increased and at the same time multi-trait methods consistently outperformed single-trait methods. These results showed that including more samples and more secondary traits that are genetically correlated with the primary trait help to improve the PA of the primary trait.

We next compared our prediction methods with the summary statistics-based prediction methods such as LDpred[8] and the recently proposed MTAG[10]. When computing PA using LDpred, we generated various candidate risk scores such as LDpred-inf, LDpred with a range of $\rho$ values (i.e. tuning parameter for the fraction of causal SNPs), and LDpred with pruning + thresholding and then determined the risk score with the best predictive capacity. For single-trait approach, we used the unadjusted PRS and the LDpred-adjusted risk scores (PRS, LDpred). For multi-trait approach, we first executed MTAG to incorporate information from summary statistics for multiple traits and then computed PA via PRS and LDpred (PRS + MTAG, LDpred + MTAG). We used the same simulation settings as previous simulations. The overall predictive performance of our methods was uniformly better than that of summary statistics-based methods in simulations (Fig. 2, Supplementary Table 1). The LDpred performed better than unadjusted PRS for all simulation settings and especially, as sample size increased, the difference of PA between LDpred and unadjusted PRS was getting larger (Fig. 2c), which can be explained by the fact that LDpred

explicitly models LD using a reference panel and estimates posterior mean effect size more accurately with large sample size than unadjusted PRS[8,34]. Starting with two traits and genetic correlation = 0.25, MTAG-based methods (PRS + MTAG, LDpred + MTAG) were similar to single-trait methods (PRS, LDpred) but MTAG-based methods performed increasingly better as number of traits increased and genetic correlation was getting larger (Fig. 2a, b). Because the MTAG estimates the trait-specific association statistics using a generalization of inverse-variance weighted meta-analysis[10], it can produce better summary data by combining multiple traits as the proportion of shared causal variants is larger with a higher genetic correlation.

We next compared our methods with GBLUP-based methods such as STGBLUP and MTGBLUP[12]. We also considered the same simulation settings as the previous ones. The overall predictive performance of STGBLUP and MTGBLUP methods was slightly better than summary statistics-based methods but uniformly worse than our methods (Fig. 2a, b). MTGBLUP utilized individual-level genotype for all traits instead of marginal SNP effects and thus it produced slightly better predictive performance than summary statistics-based methods. However, MTGBLUP performed worse than our methods because it assumed infinitesimal genetic architecture which is less appropriate in the real situation than non-infinitesimal assumption of our methods. MTGBLUP performed similar to STGBLUP with a low genetic correlation but it performed better and better as genetic correlation increased from 0.25 to 0.75 (Fig. 2a, b). Due to the limited execution time and memory, GBLUP-based methods were not feasible with large sample size ($N > 30$K) and they were excluded in the simulations for sample size (Fig. 2c).

To further explore the relationship between the predictive power of CTPR and other parameters of the data such as number

of causal SNPs, proportion of shared causal SNPs, SNP-heritability, number of traits, and sample size of summary statistics for secondary traits, we conducted the additional simulations (Supplementary Figs. 1–5). We first showed that smaller number of causal SNPs (larger per-SNP heritability) resulted in larger PA since SNPs strongly associated with the trait tend to be estimated more accurately by our prediction models (Supplementary Fig. 1). We next observed that high proportion of shared causal SNPs between two traits help to gain in PA especially at high genetic correlation (Supplementary Fig. 2). The results of Supplementary Fig. 3 explained that larger SNP-heritability of primary and/or secondary traits help PA of the primary trait. We also showed that including more secondary traits that were less genetically correlated with each other would increase the PA for the primary trait (Supplementary Fig. 4). Lastly, PA increases as sample size of the summary statistics increases and therefore all GWAS summary statistics with large sample size for related secondary traits could be used to improve the prediction model of the primary trait (Supplementary Fig. 5). A more detailed explanation is found in supplementary documents.

**Evaluation using real large-sample GWAS data from UK Biobank.** We evaluated the proposed prediction methods with real GWAS data by taking HGT as a primary trait and BMI, HIP, WST, and WHR as secondary traits. For our analysis, UK Biobank data[29–31] were used as training sets and a leave-out subset of UK Biobank or NHS/HPFS/PHS cohort data[32] as validation sets. UK Biobank data were genotyped using UK BiLEVE and UK Biobank Axiom arrays and imputed using the 1000 Genomes Phase 3 reference panels, which led to ~93 million SNPs (Supplementary Table 7). Five phenotypes of interest, HGT, BMI, HIP, WST, and WHR were measured on 456,837 individuals, who were of self-reported European ancestry. Among those individuals, we used 30,000 (30K) or 436,837 (437K) samples for training and 20,000 (20K) samples for testing. NHS/HPFS/PHS cohort data were made of 23 GWAS datasets based on three different genotype platforms and imputed via the 1000 Genomes Phase 1 reference panels (Supplementary Table 7). We again excluded individuals of non-European ancestry and had 20,769 individuals, among which 20,000 (20K) individuals were randomly selected for an independent validation set. We restricted the set of SNPs for risk prediction to Hapmap3 SNPs for comparability across the two cohorts. We prepared the overlap Hapmap3 SNPs between UK Biobank and NHS/HPFS/PHS cohort based on MAF (>0.05) and imputation $R^2$ (>0.80), which consisted of 955,842 (1M) SNPs. We divided these genotype datasets into an adequate number of files for parallel computing. We computed summary statistics for all traits which were conducted via SNPTEST with age, sex, and top 10 genotype PCs.

We performed LD score regression[35] on the GWAS summary statistics to estimate the proportion of phenotypic variance explained by all SNPs (i.e. heritability) for HGT, BMI, HIP, WST, and WHR using LDSC software[35–37]. We showed that 45.3% (s.e. = 4.1%), 24.3% (s.e. = 2.4%), 20.1% (s.e. = 2.1%), 18.7% (s.e. = 1.9%), and 14.7% (s.e. = 1.9%) of variance of HGT, BMI, HIP, WST, and WHR, respectively, can be explained by 1M SNP data in the UK Biobank. We showed that the genetic correlation between HGT and BMI was −0.130 and those between HGT and the other phenotypes were 0.304 (HIP), 0.162 (WST), and −0.112 (WHR). The phenotypic correlations between HGT and the other phenotypes were computed after adjusting for age and sex. HGT and BMI or WHR were shown to be negatively correlated ($r = -0.097, -0.059$), and HGT and HIP or WST were shown to be positively correlated ($r = 0.178, 0.088$) in the UK Biobank,

respectively (Supplementary Figs. 7 and 8). Similar patterns were found in NHS/HPFS/PHS datasets (Supplementary Fig. 7). The difference of heritability estimates between BMI and other three phenotypes, and the difference of absolute genetic correlations between HGT-BMI and HGT-other three phenotypes are not big, so we expect that the difference of PA for HGT between using BMI and other three phenotypes would not be large.

We first applied our single-trait and multi-trait approaches to predict HGT by the aid of BMI with 30K training individuals from UK Biobank. We considered age and sex-adjusted HGT as a primary trait and summary statistics for BMI as a secondary trait. To make all phenotypes on the same scale and direction, SNP effects for BMI were re-scaled to make slopes of the regression of SNP effects for HGT on SNP effects for BMI equal to 1. The performance of Lasso + CTPR was comparable to that of MCP + CTPR using either UK Biobank or NHS/HPFS/PHS as a validation set. PA evaluated in NHS/HPFS/PHS was smaller than PA tested within UK Biobank likely due to genetic heterogeneity between UK Biobank and NHS/HPFS/PHS (Fig. 3a). The results confirmed again that our multi-trait approaches improved PA compared to the single-trait approaches. With 30K training sample size, the relative gains in prediction $R^2$ of our multi-trait approaches were 11.1% for Lasso + CTPR (27.4% for MCP + CTPR) using UK Biobank and 8.7% (23.5%) using NHS/HPFS/PHS data as a validation set (Supplementary Table 9).

Next, we compared our prediction methods to the existing multi-trait methods such as MTAG and MTGBLUP with 30K training samples. For MTAG, summary statistics for HGT and BMI from the UK Biobank were used to build predictors and, for MTGBLUP, individual-level GWAS data for both traits were used. Similar to our method, the PA using MTAG and MTGBLUP decreased when testing on NHS/HPFS/PHS cohort compared to testing within UK Biobank. MTGBLUP method performed quite similar to MTAG but both methods performed much worse than CTPR using UK Biobank or NHS/HPFS/PHS cohort (Fig. 3a). Using MTAG or MTGBLUP did not show better PA compared to their corresponding single-trait methods. As expected, LDpred performed better than unadjusted PRS for both single-trait and multi-trait approaches. The PA for HGT improved from $R^2 = 11.3\%$ (LDpred + MTAG) or 11.4% (MTGBLUP) to 14.5% (MCP + CTPR) or 14.6% (Lasso + CTPR) using UK Biobank and from $R^2 = 9.0\%$ (LDpred + MTAG) or 8.5% (MTGBLUP) to 11.3% (MCP + CTPR, Lasso + CTPR) using NHS/HPFS/PHS data (Supplementary Table 9).

We then considered HIP, WST, WHR one at a time and four traits including BMI jointly as secondary traits to help prediction of HGT using our methods (Supplementary Table 10). Using our multi-trait methods, the PA of two-trait approach with HIP, WST, and WHR was slightly less than PA with BMI but PA of five-trait approach performed the best among all the analyses. This is likely because heritability for BMI is the largest among all secondary traits, while genetic correlations among secondary traits are relatively high and their genetic correlations with HGT are quite similar.

Finally, we showed that the use of full training samples ($N = 437K$) from UK Biobank can substantially improve PA compared to the smaller training samples ($N = 30K$). Similar to the 30K sample analyses, our multi-trait methods outperformed all summary statistics-based methods (LDpred, MTAG) as well as our single-trait methods (Fig. 3b). Because the GBLUP-based methods were not feasible with 437K training samples, we excluded them from full sample analyses. Especially, we found that the PA for HGT using Lasso + CTPR (42.8%) captured most of estimated SNP-heritability for HGT (45.3%) using the same UK Biobank data although our SNP-heritability estimate may be

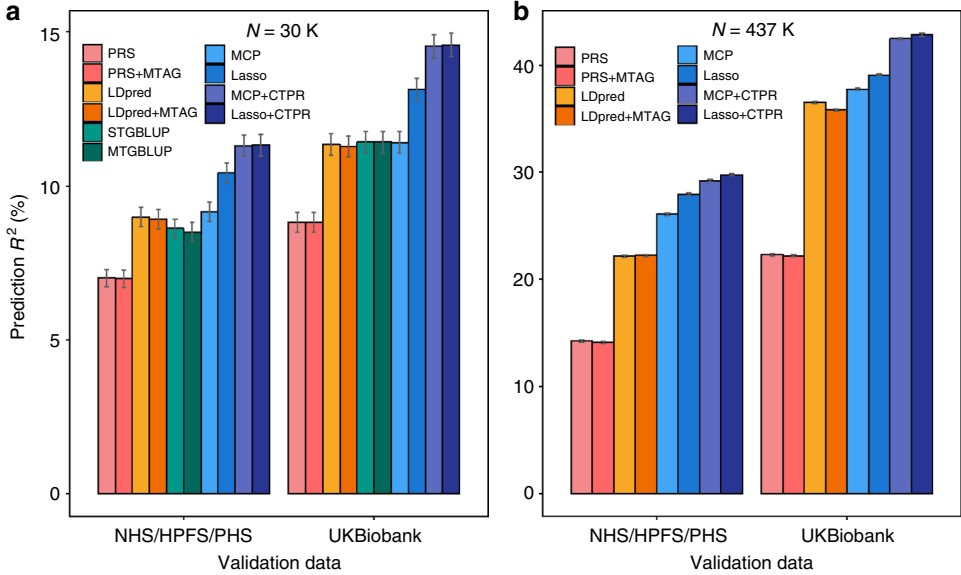

**Fig. 3** Comparisons of the predictive performance for HGT by the aid of BMI among LDpred, MTAG, STGBLUP, MTGBLUP, and CTPR using UK Biobank data ($N = 30$K or 437K) as a training set and either UK Biobank ($N = 20$K) or NHS/HPFS/PHS cohort ($N = 20$K) as a validation set. We considered human height (HGT) as the primary trait and body mass index (BMI) as the secondary trait. Prediction $R^2$ were computed with two different validation sets (UK Biobank, NHS/HPFS/PHS data) and two different training sample sizes (30K, 437K) using summary-based prediction methods (LDpred, MTAG), GBLUP (STGBLUP, MTGBLUP), and the proposed prediction methods (MCP/MCP + CTPR, Lasso/Lasso + CTPR). Clearly, our multi-trait methods (Lasso + CTPR, MCP + CTPR) generated better predictive performance than our single-trait methods and the existing prediction methods with different training sample sizes and different validation sets. All error bars represent standard errors of prediction $R^2$. **a** Real GWAS analysis results with 30K training samples for prediction $R^2$ of HGT by the aid of BMI. **b** Real GWAS analysis results with 437K training samples for prediction $R^2$ of HGT by the aid of BMI

underestimated (because current SNP-heritability estimate ranges from 45% (ref. [38]) to 54% (ref. [39])) and thus 45.3% may not be the upper limit of the PA[40]. The PA for HGT improved from $R^2 = 35.8\%$ (LDpred + MTAG) to 42.5% (MCP + CTPR) or 42.8% (Lasso + CTPR) using UK Biobank and from $R^2 = 22.3\%$ (LDpred + MTAG) to 29.2% (MCP + CTPR) or 29.8% (Lasso + CTPR) using NHS/HPFS/PHS data as a validation set (Supplementary Table 9).

The scatter plots for actual HGT vs. predicted HGT of the 20K testing samples from NHS/HPFS/PHS or UK Biobank provided practical insight on how differently our multi-trait method (Lasso + CTPR) and the summary statistics-based multi-trait method (LDpred + MTAG) generate the predicted outcomes, using 30K or 437K training samples, respectively (Fig. 4). Two distinguishable groups in each scatter plot represented different sex groups (i.e. men and women groups). The scatter plots clearly showed that our multi-trait method (Lasso + CTPR) produced much better predicted HGT than the other method (LDpred + MTAG). Also, the use of the same cohort as a testing set helped to generate better predicted values. Furthermore, more samples enabled Lasso + CTPR and LDpred + MTAG to show better predictive performance (437 K sample analyses vs. 30 K sample analyses). We found that PA of men was always smaller than PA of women, which can be explained by the fact that the training sample size for women is larger than men and heritability for women HGT is larger than heritability for men HGT in UK Biobank.

## Discussion
We have developed a novel statistical framework for cross-trait penalized regression for polygenic risk prediction, CTPR, using individual-level genotype data and GWAS summary statistics, and shown that the CTPR can utilize biobank-based GWAS data for multi-trait risk prediction in a computationally efficient way

based on a new MPI algorithm (Fig. 1). We have further demonstrated in extensive simulations and real GWAS data analyses from UK Biobank and NHS/HPFS/PHS cohort that the proposed multi-trait methods (Lasso + CTPR or MCP + CTPR) produced better prediction performance over existing single-trait and multi-trait methods such as MTAG and MTGBLUP. With 30K training samples from UK Biobank, the PA of Lasso + CTPR for human height by aid of BMI improved 29.2% over MTAG and 28.1% over MTGBLUP and with 437 K training samples, the PA of Lasso + CTPR improved 18.9% over MTAG (Fig. 3).

The CTPR outperforms the existing multi-trait prediction methods such as MTAG and MTGBLUP for two main reasons. First, the CTPR fits all SNPs simultaneously using penalized regression, which can produce more precise estimates of all SNPs than MTAG because MTAG utilizes only marginal SNP effects for multiple traits. Although marginal SNP effects can be re-analyzed to account for LD through approximate summary statistic BLUP predictors (SBLUP) or approximate mixture model predictors (LDpred), summary statistics-based methods perform worse than GBLUP-based methods with individual-level data in general[9,34]. The predictive performance of wMT-SBLUP is also mostly worse than MTGBLUP because wMT-SBLUP is considered as a natural extension of MTGBLUP to summary statistics[9]. Based on our simulation and real data analyses, our CTPR outperformed MTGBLUP and consequently wMT-SBLUP. Second, the CTPR takes advantage of non-infinitesimal genetic architecture, which enables a gain in PA relative to the existing infinitesimal model-based methods such as MTGBLUP. The CTPR utilizes shrinkage methods to estimate SNP effects more accurately than MTGBLUP and thus achieves better estimates for non-causal genetic effects and better prediction than MTGBLUP under non-infinitesimal genetic architecture. Wood et al.[41] showed that the genetic architecture for human height is

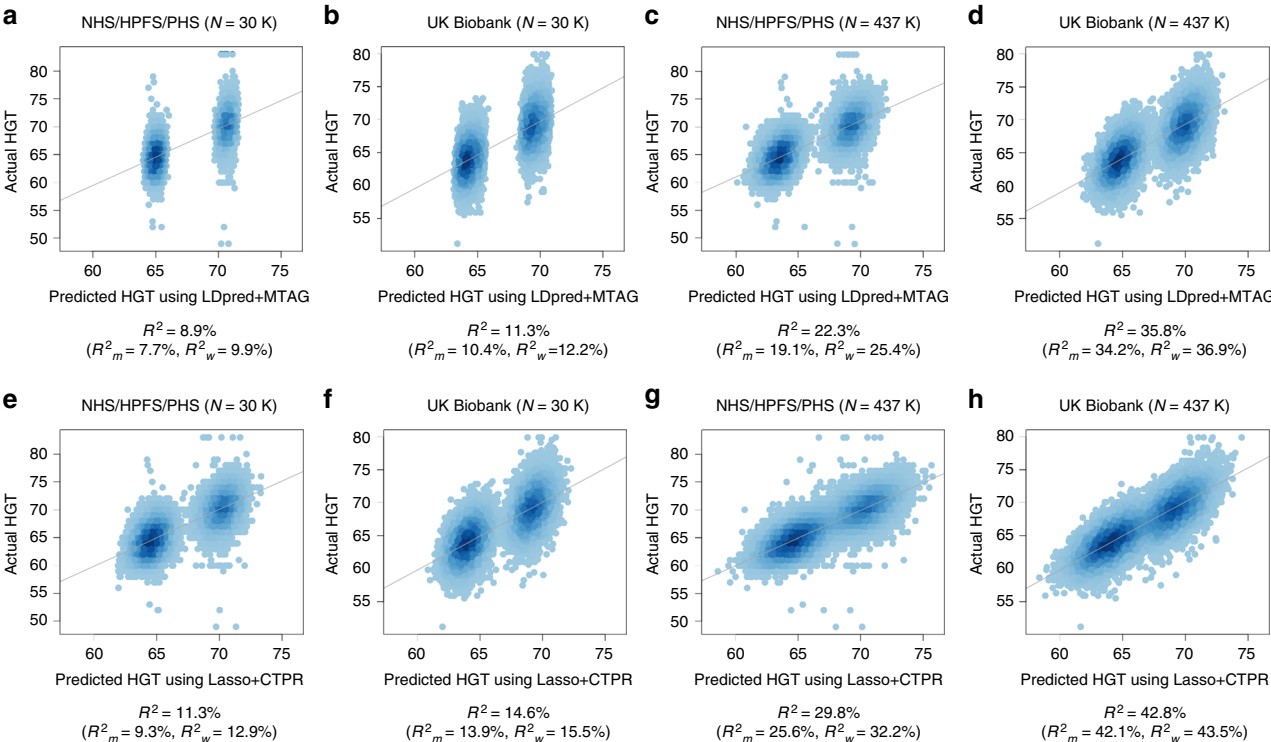

**Fig. 4** Scatter plots of actual HGT vs. predicted HGT of 20K testing samples from either NHS/HPFS/PHS cohort or UK Biobank which were estimated by 30K or 437K training samples from UK Biobank using LDpred + MTAG and Lasso + CTPR. We provided prediction $R^2$ as well as sex-specific prediction $R^2$ ($R^2_m$: PA for men, $R^2_w$: PA for women). We showed scatter plots of actual HGT vs. predicted HGT of 20K testing samples from NHS/HPFS/PHS cohort or UK Biobank data to give practical insight on how differently Lasso + CTPR and LDpred + MTAG generate predicted outcomes using 30K or 437K training samples from UK Biobank. Prediction $R^2$ for men is constantly smaller than prediction $R^2$ for women and the use of the different cohort as a validation set produced worse predicted values than the use of the same cohort. Lasso + CTPR generated better predicted HGT than LDpred + MTAG for all analyses

characterized by a considerably large but finite number (e.g. thousands) of causal SNPs. Recently, the proportion of underlying susceptibility SNPs for 32 complex traits were estimated using GWAS summary statistics and external LD information based on a two or three component normal-mixture model[42], and 1.48% of SNPs from GIANT consortium ($N = 253K$) were expected as causal for human height. Most complex traits and disease phenotypes from a variety of sources were expected to have a smaller proportion of causal SNPs than human height (0.15–1.40%). Since the genetic architecture underlying these broad ranges of phenotypes have proportion of causal SNPs far from the infinitesimal model, we expect that our multi-trait methods will perform well for many phenotypes of interest. Also, we found Lasso-based methods perform increasingly better than Ridge/GBLUP-based methods as training sample size increases at the same proportion of causal variants in our simulation studies (Supplementary Figs. 9 and 10). Furthermore, our method runs through the proposed distributed memory MPI algorithm and thus it is computationally feasible for biobank-based large-scale GWAS data but GBLUP-based method is impractical because it requires the estimation of GRM matrix, which become computationally prohibitive as the sample size increases.

For a cross-trait penalty, Li and Li[27] and Kim and Xing[28] proposed to use $L_1$ norm of the difference between two coefficients[28] as well as $L_2$ norm. Indicator variables for SNP effects can be even employed. For this paper, we selected a Laplacian quadratic penalty to incorporate related traits because the proposed cross-trait penalty function is easier to obtain the close form solution for a single coefficient and computationally more

efficient than other penalty forms. Furthermore, it imposes the smoothness over coefficients to remedy the disadvantage of Lasso or MCP penalties, which is one of the advantages of the elastic net[20].

Our cross-trait penalty term directly compares SNP effects of two phenotypes and does not scale SNP effects during the estimation step for genetic prediction. Therefore, it becomes an important issue to make all phenotypic values on the same scale and direction. Furthermore, our cross-trait penalty can utilize fixed GWAS summary statistics whose phenotypes and genotypes are rarely available. Since we generally have no information on phenotypes and genotypes for such summary statistics, it is necessary to implement an appropriate method to handle this issue. To make all phenotypic values on the same scale and direction, we first compute all marginal SNP effects via simple linear regression and then re-scale the secondary trait to make the slope for SNP effects of the primary trait regressed on SNP effects of the secondary traits equal to 1. This procedure ensures all SNP effects for all traits on the same scale as well as the same direction. For example, HGT and BMI are negatively correlated and thus the slope for beta coefficients of HGT regressed on beta coefficients of BMI is negative. The CTPR utilizes BMI multiplied by the slope for HGT~BMI regression as a secondary trait and it finally uses positively correlated BMI on the same scale. This has been implemented in the software CTPR (Cross-Trait Penalized Regression). The complete pipeline used to install and execute the CTPR, including required computing resources, input file formats, and optional parameters, is provided on GitHub (see the URL below). On top of that, we are implementing the pipeline on

public cloud platforms (e.g. Amazon web services, Google cloud platform, and Microsoft Azure) such that users do not need to compile and install the pipeline by themselves.

The adjacency coefficients in our cross-trait penalty function provide a convenient way to incorporate prior knowledge about the relation between the phenotypes. Here, we assumed no prior knowledge and thus set all adjacency coefficients to 1. To improve PA, we can use more appropriate values such as the cross-trait heritability estimates from literature on large-scale GWAS studies, which quantifies the genetic contribution to the covariance between two traits. Alternatively, we can consider SNP-specific adjacency measures based on their correlation coefficients with the respective phenotypes. Also, SNP-set specific adjacency measures can be considered based on cross-trait heritability computed from a specific SNP-set when functional information on SNPs are available. All SNPs are divided into multiple SNP-sets based on functional information and then each cross-trait heritability estimate for each SNP-set can be computed and used as adjacency coefficients.

Although our prediction methods provide powerful predictive performance, they have some limitations. First, the proposed methods require more computation time than summary statistics-based methods such as LDpred and MTAG although our methods outperform them. Novel MPI algorithm were thus implemented using parallel computing to reduce computation time and utilize computing resource effectively. This algorithm can compute exact (i.e. same as obtained by using the original coordinate decent algorithm) SNP effects as well as approximate ones and both are remarkably similar once the convergence is achieved. Interestingly, the computation time of our method increases only linearly with the sample size, number of SNPs, and number of core-groups and also larger sample size enables us to specify smaller number of core-groups, which makes it feasible for use in future large-sample GWAS analyses. Second, the current models described here were designed only for continuous traits. In principle, the current methods can be directly applied to binary traits by treating them as continuous traits with values of 0 and 1 although the performance on binary traits still need to be evaluated using simulation and real GWAS data. In general, the CTPR can suffer miscalibration for binary traits and specifically miscalibration takes place when minor allele count multiplied by the proportion of cases is small. Thus, we recommend to use binary traits with a case proportion of at least 10% and SNPs with MAF > 0.1% in biobank-based GWAS data[16].

In conclusion, the proposed cross-trait prediction method described here is a powerful and practical tool to predict a trait of interest by incorporating other related traits using large-scale GWAS data. It can utilize GWAS summary statistics from many related secondary traits for better prediction. We expect our proposed method will become more useful as larger sample GWAS data become accessible via either individual-level data or summary statistics.

## Methods

**UK Biobank data.** UK Biobank is a large-scale national health resource of over 500,000 individuals from across the United Kingdom (UK). Study participants were invited to 1 of 22 centers across the UK between 2006 and 2010, described in detail elsewhere[29]. Over 500,000 subjects with their phenotypes in the UK Biobank, ~488,377 subjects were genotyped at ~800,000 SNPs using the UK BiLEVE and UK Biobank Axiom Arrays from Affymetrix. Pre-phasing was carried out using a modified version of the SHAPEIT2 (ref. [43]) and imputation was performed with the UK10K and the 1000 Genomes Phase 3 reference panels using IMPUTE2 (ref. [44]), which resulted in ~93 million SNPs (Supplementary Table 7). With MAF > 0.05, imputation $R^2$ > 0.8, and $P$(HWE) > $10^{-10}$, we finally have 5,498,274 SNPs. Genotyping, quality control for SNPs and samples, and imputation procedures are described in detail here[30,31]. Individuals of non-European ancestry were excluded to be consistent with NHS/HPFS/PHS cohort data, which results in 456,837

individuals. For simulation studies and real GWAS analyses, we prepared 30,000 (30K) and 436,837 (437K) individuals for a training set and the remaining 20,000 (20K) individuals for a validation set.

**NHS/HPFS/PHS cohort data.** We combined 23 GWAS datasets from Nurses' Health Study I, II (NHS, NHS2), Health Professionals Follow-up Study (HPFS), and Physicians' Health Study (PHS) into three compiled datasets based on their genotype platform types: Affymetrix, Illumina HumanHap, and Illumina OmniExpress. We eliminated any SNPs that were not in all studies or with high missingness (>0.05), which led to 668,283 SNPs in the Affymetrix, 459,999 SNPs in the Illumina HumanHap, and 565,810 SNPs in the Illumina OmniExpress dataset. Based on a pairwise identity by descent (IBD) analysis for each combined dataset, we removed individuals that are duplicates (or identical twins) or flagged for unexpected duplicates with the different cohort IDs but pairwise genotype concordance rate >0.99. Related individuals (full sibs, half sibs) were not removed in each dataset. This resulted in 8065 individuals in the Affymetrix dataset, 6787 individuals in the Illumina HumanHap, and 5,917 individuals in the OmniExpress, the total number of which is 20,769. For imputation, the 1000 Genomes Project Phase I Integrated Released Version 3 Haplotypes excluding monomorphic and singleton sites (2010–11 data freeze, 2012-03-14 haplotypes) were used as reference panel. Genotypes on each chromosome were first split into chunks and each chunk of chromosome was pre-phased using MACH[45] (v.1.0.18.c). Imputation was performed using Minimac[46] (v.2012-08-15), which finally led to ~31 million SNPs for each dataset (Supplementary Table 7). We obtained 5,354,676 SNPs after removing all SNPs with MAF <0.05 and imputation $R^2$ < 0.8. For the simulation studies, we randomly chose 20,700 individuals in the merged dataset, among which 14,800 individuals in the Affymetrix and Illumina datasets were used as a training set and 5900 individuals in the OmniExpress dataset were used as a validation set. For real GWAS analysis, we selected 20,000 (20K) individuals to use an independent validation set (Supplementary Table 8).

**Prediction model.** Suppose we have $K$ traits with $n_k$ ($k = 1,…,K$) independent samples for each trait. We assume the first trait to be the primary one we wish to predict. Denote $y_{ki}$ as the phenotypic value and $x_{kij}$ as the genotype for the $j$th ($j = 1,…,P$) SNP of the $i$th sample with the $k$th trait. Assume $y_{ki} = \alpha_k + \sum_{j=1}^{P} x_{kij}\beta_{kj} + \epsilon_{ki}$ and $\epsilon_{ki} \sim N(0, \sigma_k^2)$, where $\alpha_k$ is the intercept and $\beta_{kj}$ is the coefficient for $j$th SNP and $k$th trait. The coefficients can be estimated using the following least-squares method:

$$\widehat{\boldsymbol{\beta}} = \text{argmin}_{\beta} + \sum_{k=1}^{K} \sum_{i=1}^{n_k} \frac{1}{2n_k} \left( y_{ki} - \alpha_k - \sum_{j=1}^{p} x_{kij}\beta_{kj} \right)^2. \tag{1}$$

Note the solution of (1) is equivalent to minimizing the objective function for each trait separately. Usually $P$ is larger than $n_k$.

**Sparsity penalty.** To avoid overfitting, we add to (1) a sparsity penalty $p_{\lambda_1}^{\text{sp}}(\boldsymbol{\beta})$ to induce a sparse solution. For $p_{\lambda_1}^{\text{sp}}(\boldsymbol{\beta})$, we investigate two possibilities, the Lasso[18] and the MCP[22]. The Lasso penalty, defined as $p_{\lambda_1}^{\text{sp}}(\boldsymbol{\beta}) = \lambda_1 \sum_{k=1}^{K} \sum_{j=1}^{P} \left| \beta_{kj} \right|$, provides an efficient way to induce sparsity due to its non-differentiability at 0. Under certain regularity conditions, Lasso achieves variable selection consistency. Its convexity makes optimization very straightforward but at the same time produces biased estimates due to shrinkage of the non-zero coefficients. The MCP, though not convex, provides the convexity of the penalized loss in sparse regions to the greatest extent while trying to preserve the variable selection and unbiasedness features. In other words, MCP imposes as much penalty as possible on small coefficients but less or no penalty on large coefficients. It is defined as $p_{\lambda_1,\gamma}^{\text{sp}}(\boldsymbol{\beta}) = \lambda_1 \sum_{k=1}^{K} \sum_{j=1}^{P} \int_0^{|\beta_{kj}|} \left( 1 - \frac{x}{\gamma\lambda_1} \right)^+ \mathrm{d}x$, where $(.)^+$ is a function which sets negative values to 0 and $\gamma$ is an additional regularization parameter with large value providing smoother estimators but larger bias and less accurate variable selection. When $\gamma \to \infty$, it converges to the Lasso. In practice, it can be set to a constant to reduce computational complexity.

**Cross-trait penalty.** If we assume the phenotypes are correlated and affected by some common causal SNPs with similar effects, we can add to (1) another cross-trait penalty $p_{\lambda_2}^{\text{ctp}}(\boldsymbol{\beta})$ to induce certain smoothness of the coefficients with respect to the relationship between two traits. For $p_{\lambda_2}^{\text{ctp}}(\boldsymbol{\beta})$, we take it to be Laplacian quadratic penalty, which can be defined as $p_{\lambda_2}^{\text{ctp}}(\boldsymbol{\beta}) = \frac{\lambda_2}{2} \sum_{k \neq k'}^{K} \sum_{j=1}^{P} a_{kk'j} \left( \beta_{kj} - \beta_{k'j} \right)^2$ where $a_{kk'j}$ is the adjacency coefficient, which can be used for incorporating the prior knowledge on the similarity of the coefficient between trait $k$ and $k'$ for SNP $j$. If we have $M$ additional traits for which only summary statistics, not individual-level genotypes, are available for all coefficients, these traits can be also used through adding additional quadratic penalties. The modified cross-trait penalty

is defined as

$$p_{\lambda_2}^{ctp}(\boldsymbol{\beta}) = \frac{\lambda_2}{2}\left\{\sum_{k\neq k'}^{K}\sum_{j=1}^{P} a_{kk'j}\left(\beta_{kj} - \beta_{k'j}\right)^2 + \sum_{k=1}^{K}\sum_{m=1}^{M}\sum_{j=1}^{P} a_{k(K+m)j}\left(\beta_{kj} - \hat{s}_{mj}\right)^2\right\},$$

(2)

where $\hat{s}_{mj}$ is the fixed summary statistics for additional traits. We can define the adjacency coefficient to incorporate prior knowledge on how the phenotypes are related but we set $a_{kk'j} = 1$ for all SNPs in this study, which assumes no prior knowledge. Finally, the sparse and smoothed estimate of $\boldsymbol{\beta}$ can be estimated by

$$\hat{\boldsymbol{\beta}} = \arg\min_\beta \sum_{k=1}^{K}\sum_{i=1}^{n_k}\frac{1}{2n_k}\left(y_{ki} - \alpha_k - \sum_{j=1}^{P} x_{kij}\beta_{kj}\right)^2 + p_{\lambda_1}^{sp}(\boldsymbol{\beta}) + p_{\lambda_2}^{ctp}(\boldsymbol{\beta}).$$

(3)

**Coordinate descent algorithm.** We first standardize the data so that $\sum_{i=1}^{n_k} x_{kij}^2 = n_k$, $\sum_{i=1}^{n_k} x_{kij} = 0$, and $\sum_{i=1}^{n_k} y_{ki} = 0$, so that the intercept will not be included in the model. Coordinate descent algorithm[47,48], which optimizes over one coefficient at a time with other coefficients fixed, can be used for obtaining the solution of (3). Because there is a closed form solution for optimization over a single coefficient, the algorithm is very computationally efficient and can handle a large number of SNPs. Suppose $\tilde{\boldsymbol{\beta}}$ are current estimates and we want to optimize over $\beta_{k'j'}$. The univariate optimization over $\beta_{k'j'}$ becomes

$$\hat{\beta}_{k'j'} = \arg\min_{\beta_{k'j'}}\frac{1}{2}\left(\beta_{k'j'} - \tilde{b}_{k'j'}\right)^2 + p_{\lambda_1}^{sp}\left(\beta_{k'j'}\right) + \frac{\lambda_2}{2}d_{k'j'}\beta_{k'j'}^2, \tilde{b}_{k'j'} = \tilde{\zeta}_{k'j'} + \tilde{\xi}_{k'j'},$$

$$\tilde{\zeta}_{k'j'} = \left(\sum_{i=1}^{n_k} x_{k'ij'}\tilde{r}_{k'ij'}\right)/n'_k, \tilde{r}_{k'ij'} = y_{k'i} - \sum_{j\neq j'} x_{k'ij}\tilde{\beta}_{k'j}, \tilde{\xi}_{k'j'} =$$

$$\lambda_2\left(\sum_{k\neq k'} a_{kk'j}\tilde{\beta}_{kj} + \sum_{k=1}^{K}\sum_{m=1}^{M} a_{k(K+m)j}\hat{s}_{mj'}\right), d_{k'j'} = \sum_{k\neq k'} a_{kk'j'} \text{ and}$$

$$p_{\lambda_1}^{sp}\left(\beta_{k'j'}\right) = \begin{cases} \lambda_1\left|\beta_{k'j'}\right| & \text{if the penalty is Lasso} \\ \lambda_1\int_0^{|\beta_{k'j'}|}\left(1 - \frac{x}{\gamma\lambda_1}\right)^+ dx & \text{if the penalty is MCP} \end{cases}.$$ For Lasso penalty, the solution is given by

$$\hat{\beta}_{k'j'} = sgn\left(\tilde{b}_{k'j'}\right)\frac{\left(|\tilde{b}_{k'j'}| - \lambda_1\right)^+}{1 + \lambda_2 d_{k'j'}},$$

(4)

where $sgn(.)$ is the sign function. For MCP penalty, the solution is given by

$$\hat{\beta}_{k'j'} = \begin{cases} sgn\left(\tilde{b}_{k'j'}\right)\frac{\gamma\left(|\tilde{b}_{k'j'}| - \lambda_1\right)^+}{\gamma\left(1 + \lambda_2 d_{k'j'}\right) - 1} & \text{if } \left|\tilde{b}_{k'j'}\right| \leq \gamma\lambda_1\left(1 + \lambda_2 d_{k'j'}\right), \\ \frac{\tilde{b}_{k'j'}}{1 + \lambda_2 d_{k'j'}} & \text{if } \left|\tilde{b}_{k'j'}\right| > \gamma\lambda_1\left(1 + \lambda_2 d_{k'j'}\right) \end{cases}.$$

(5)

It is easy to see that when $\gamma \to \infty$, $\left|\tilde{b}_{k'j'}\right|$ will be much smaller than $\gamma\lambda_1\left(1 + \lambda_2 d_{k'j'}\right)$ and the Eq. (5) will converge to the Eq. (4). The coordinate descent algorithm cycles through all the coefficients for all the traits until a certain convergence criterion is reached such as the largest relative change of the coefficients from two consecutive cycles is less than $10^{-3}$. Let $N = \sum_{k=1}^{K} n_k$, then the computational complexity of above algorithm is $O(NP^2)$. To reduce the computational complexity to $O(NP)$, the residuals can be updated using

$$\tilde{r}_{k'i(j'+1)} = \tilde{r}_{k'ij'} - \hat{\beta}_{k'j'}x_{k'ij'} + \tilde{\beta}_{k'(j'+1)}x_{k'i(j'+1)}.$$

(6)

Considerable speedup can be achieved by organizing the iterations around the features with non-zero coefficients (active sets). After a complete cycle through all the coefficients, we iterate on only the active set till convergence. The process repeats until another complete cycle does not change the active set. Given a fixed value of $\lambda_2$, we compute the solutions for a decreasing sequence of values for $\lambda_1$ on the log scale, starting at the smallest value $\lambda_1^{max}$ that produces the sparsest model $\left(\hat{\boldsymbol{\beta}} = 0\right)$. $\lambda_1^{max}$ can be estimated as $\max_{k,j}\left|\sum_i x_{kij}y_{ki}\right|/n_k$. Apart from giving us a path of solutions, this scheme exploits warm starts, and leads to a more stable algorithm.

**Tuning parameters.** To select the tuning parameter $\lambda_1$ and $\lambda_2$, we use $n$-fold cross-validation. Since our focus is to improve the prediction of the primary phenotype, we only divide the dataset for the primary phenotype into $n$ folds, each fold will be used as the validation dataset and the remaining as the training dataset. The datasets for secondary phenotypes are not divided and will be fully used. However, if some individuals have data on both the primary phenotype and secondary phenotypes, the secondary ones are also divided into $n$ folds for those individuals to avoid overfitting. We then run the penalized least-squares method on the training dataset and obtain the MSE on the validation dataset. We select the values for $\lambda_1$ and $\lambda_2$ that minimize the averaged MSE over the $n$ folds. Though $\gamma$ is also

tunable, we could fix $\gamma$ to reduce computational cost while still achieving good performance. We use $\gamma = 3.0$ following the author's suggestion[22].

**Novel MPI algorithm for biobank-based GWAS data.** Recent biobank-based GWAS data contain several millions of SNPs in hundreds of thousands of individuals. Due to limited memory and computing resources, it may not be feasible to update the coefficients for all SNPs simultaneously. Alternatively, we propose to divide SNPs into multiple subgroups and allocate each subgroup to an MPI core for parallel computing. Each core updates only parameters in the subgroup with the remaining coefficients transferred from other cores. Because MPI allows for communication between different cores, the data of all cores are synchronized at each estimation step. A detailed description of MPI algorithm is below.

Whole-genome SNP data are divided into non-overlapping subgroups which contain the fixed number of SNPs in low LD. We consider $q$ subgroups where SNPs within the same subgroup are independent. Each MPI core is assigned to one of $q$ subgroups and then all cores run simultaneously at each estimation step while the residuals of the specific subgroup are updated in a sequential way (see Eq. (6)). Now, we consider both extreme cases, where the number of subgroup $q$ is either 1 or $P$ (i.e. the number of all SNPs). When $q = 1$, we have one subgroup and the coefficients for all SNPs are updated sequentially. When $q = P$, the number of subgroups equals the number of SNPs and thus all SNPs are updated concurrently. In the former case, there is no gain in computation time and the latter case may lead to some convergence issues. There is a need to balance between them. We propose to use another group, i.e. $s$ core-groups ($s \leq q$), each of which contains several subgroups (see Fig. 1). All cores in each core-group run simultaneously at each step, keeping all cores in other core-groups waiting till finish. In this way, coefficients of the specific core-group are updated in a consecutive order to avoid convergence problem as well as to use more computing cores.

More detailed, we consider two sequence of index sets, such as $q$ ($\leq P$) mutually exclusive index sets $\{\boldsymbol{I}^1, \boldsymbol{I}^2, ..., \boldsymbol{I}^q\}$ where $\bigcup_{h=1}^{q}\boldsymbol{I}^h = \{1, ..., P\}$ and $\boldsymbol{I}^h \cap \boldsymbol{I}^{h'} = \varnothing$, $h \neq h'$, and $s$ ($\leq q$) ordered index sets $\{\boldsymbol{A}^1, \boldsymbol{A}^2, ..., \boldsymbol{A}^s\}$ where $\bigcup_{f=1}^{s}\boldsymbol{A}^f = \{1, ..., q\}$ and $a^1 < a^2 < ... < a^s$ for any element $a \in \boldsymbol{A}^f$. We partition the parameters $\tilde{\boldsymbol{\beta}}$ into $q$ subvectors, $\tilde{\boldsymbol{\beta}} = \left(\tilde{\boldsymbol{\beta}}^{1(1)}, ..., \tilde{\boldsymbol{\beta}}^{s(q)}\right)$ where $\tilde{\boldsymbol{\beta}}^{f(h)} = \left\{\hat{\beta}_{kj}|j \in \boldsymbol{I}^h, h \in \boldsymbol{A}^f\right\}$ is the current estimates corresponding to the $h$th subgroup in the $f$th core-group. In Eqs (4) and (5), the solutions for both the Lasso and MCP penalties contain $\tilde{b}_{k'j'}$ which can be expressed as, for $h$th subgroup in the $f$ th core-group,

$$\tilde{b}_{k'j'}^{f(h)} = \frac{1}{n_k}\sum_{i=1}^{n_k} x_{k'ij}\tilde{r}_{k'ij'}^{f(h)} + \lambda_2\left(\sum_{k\neq k'} a_{kk'j}\tilde{\beta}_{kj}^{f(h)} + \sum_{k=1}^{K}\sum_{m=1}^{M} a_{k(K+m)j}\hat{s}_{mj'}^{f(h)}\right),$$

where

$$\tilde{r}_{k'ij'}^{f(h)} = y_{k'i} - \sum_{j\neq j', j \in \boldsymbol{I}^h} x_{k'ij}\tilde{\beta}_{k'j}^{f(h)} - \sum_{f'<f}\sum_{h'\in\boldsymbol{A}^{f'}}\sum_{j\in\boldsymbol{I}^{h'}} x_{k'ij}\hat{\beta}_{k'j}^{f'(h')} - \sum_{f'>f}\sum_{h'\in\boldsymbol{A}^{f'}}\sum_{j\in\boldsymbol{I}^{h'}} x_{k'ij}\tilde{\beta}_{k'j}^{f'(h')}.$$

(7)

Since the last two terms of $\tilde{r}_{k'ij'}^{f(h)}$ are transferred from other cores, the coefficients for SNPs within the $h$th subgroup in the $f$th core-group are easily estimated. When $s = 1$, the coefficients in each subgroup are updated sequentially and when $s = q$, the proposed MPI algorithm becomes the original coordinate descent algorithm but taking $1/q$ required memory on each core.

**Model for simulation.** We simulated $K$ traits, each with $n_k$ samples ($k = 1, ..., K$). We selected $P$ SNPs and set the number of causal SNPs to $C$ ($C \leq P$) which were randomly selected. To generate the phenotype, we used the following linear model: $y_{ki} = \sum_{j=1}^{P} x_{kij}\beta_{kj} + \epsilon_{ki}, \epsilon_{ki} \sim N(0, \sigma_k^2), k = 1, ..., K$. For considering the genetic correlation among $K$ traits, $\boldsymbol{\beta}_j = (\beta_{1j}, ..., \beta_{Kj})$ were generated from the following multivariate normal distribution: $\boldsymbol{\beta}_j \sim N_K(\boldsymbol{0}, \boldsymbol{D})$ if $j$th SNP is causal, $\boldsymbol{\beta}_j = \boldsymbol{0}$ if $j$th SNP is non-causal where $\boldsymbol{D}$ is a $K \times K$ covariance matrix with $(l, m)$ element denoted by $d_{lm}$. If we set $var(y_{ki}) = 1$, the heritability for $k$th trait is $h_k^2 = var\left(\sum_{j=1}^{P} x_{kij}\beta_{kj}\right)$. Assuming $x_{kij}$ and $\beta_{kj}$ are independent to each other and $x_{kij}$ are standardized, $h_k^2 = C \cdot var\left(\beta_{kj}\right) = C \cdot d_{kk}$ and hence $d_{kk} = h_k^2/C$. The covariance between two phenotypes, $d_{lm}(l \neq m)$ can be calculated using $d_{lm} = \rho_{lm}\sqrt{d_{ll}}\sqrt{d_{mm}}$ where $\rho_{lm}$ is the correlation between $\beta_{lj}$ and $\beta_{mj}$. Since $var(y_{ki}) = 1$, the random error is $\sigma_k^2 = 1 - h_k^2$. In this model, SNP-heritability is defined as variance explained by the causal SNPs because beta coefficients for multiple traits are generated from multivariate normal distribution only for causal SNPs and other beta coefficients are set to zero. The covariance matrix for multivariate normal distribution are designed with number of causal SNPs, desired SNP-heritability, and genetic correlation among traits.

**Family structure and cryptic relatedness in UK Biobank.** The UK Biobank data contains related samples, including twins, parent–offspring and sibling pairs. A total of 142,726 training samples (32.7%) have at least one relative (third degree or closer) in the cohort, which leads to 314,111 unrelated training samples. Also, a

total of 6315 testing samples (31.6%) among 20,000 are inferred to be related with someone in the full cohort. In order to check how related samples between training and testing sets from UK Biobank have impact on potential inflation of PA, we re-computed PA using only unrelated testing samples ($N = 13,685$) instead of all testing samples and then conducted subsampling hypothesis tests[49]. The unrelated testing samples ($N = 13,685$) have no relatives (third degree or closer) in the cohort and hence there should be no relatives between training and testing sets. With the same 437K training samples, we observed small reduction in PA using unrelated testing samples but difference was not statistically significant based subsampling $p$-values (change in PA from previous to current results ranged from $+0.0023$ to $-0.005$, min $p$-value $>0.422$ for $H_0$: difference $= 0$, see Supplementary Table 12). For further comparisons, we randomly selected the same number of testing samples ($N = 13,685$) from NHS/HPFS/PHS cohort and re-computed PA using only these samples. The results were still similar to our previous results and NHS/HPFS/PHS samples still showed smaller PA than UK Biobank testing, likely due to genetic heterogeneity between UK and US samples, and UK Biobank samples are likely environmentally more homogeneous than samples within NHS/HPFS/PHS cohort. Furthermore, the recent paper[19] investigated the impact of relatives in UK Biobank on PA and concluded that there was no discernable difference in prediction results between using a training set drawn from the set of kinship-filtered samples and samples from the unfiltered set. Due to notably large number of related samples (>30%), removing these samples will result in a nontrivial decease in training sample size and PA. Because we found the statistically non-significant impact of related samples in UK Biobank on PA, we utilized full UK Biobank samples for our analyses.

**URLs**. The URLs presented in the paper are as follows. UK Biobank: http://www.ukbiobank.ac.uk/, LDpred: https://github.com/bvilhjal/ldpred, MTGBLUP: https://github.com/uqrmaie1/mtgblup, MTAG: https://github.com/omeed-maghzian/mtag/.

**Reporting Summary**. Further information on experimental design is available in the Nature Research Reporting Summary linked to this article.

**Code availability**. The CTPR software is available at https://github.com/wonilchung/CTPR. Data analysis codes are available from the corresponding author upon reasonable request.

## Data availability

All predictors for human height by aid of BMI with 437K training samples from UK Biobank using PRS, LDpred, MCP, Lasso, PRS + MTAG, LDpred + MTAG, MCP + CTPR, and Lasso + CTPR are available for academic uses at http://lianglab.rc.fas.harvard.edu/CTPR/. All other data are available from the corresponding author upon reasonable request.

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

## Acknowledgements

We appreciate the individuals who participated in the study. The work described in this paper was funded by R01 GM105857-01A1. The Nurses' Health Study Genome-Wide Association Study database was by the following grants from the National Institutes of Health: P01CA87969, P01CA055075, P01DK070756, U01HG004728, UM1CA186107, UM1CA176726, R01CA49449, R01CA50385, R01CA67262, R01CA131332, R01HL034594, R01HL088521, R01HL35464, R01HL116854, R01EY015473, R01EY022305, P30EY014104, R03DC013373, and R03CA165131. This research has been conducted using the UK Biobank Resource (application number 16549).

## Author contributions

W.C., J.C., and L.L. conceived and designed the experiments. W.C. and L.L. performed the experiments and analyzed the data. W.C., C.T., S.L., Z.Z., P.-R.L., and P.K. generated data, materials and analysis tools. W.C., J.C., and L.L. wrote the paper. All authors reviewed and revised the manuscript.

## Additional information

**Competing interests:** The authors declare no competing interests.

