## [Peer Review File · Nature Communications]

Reviewers' comments:

Reviewer #1 (Remarks to the Author):

This well-written manuscript proposes an approach to improve polygenic risk prediction in GWAS studies of a continuous phenotype by leveraging possible shared genetic effects with secondary phenotype(s) correlated with the primary phenotype. To do this, the authors start with a penalized regression procedure (either Lasso or MCP) for each phenotype and then incorporate an additional cross-trait penalty function using a Laplacian quadratic penalty that is a function of the squared difference in SNP beta coefficients between the primary and secondary phenotype. A novel computational algorithm is then utilized to allow the method to scale to standard GWAS projects. The method is illustrated using both simulated data and a GWAS study of height that leverages additional cross-trait information from independent genetic studies of age of menarche and BMI. The statistical algorithm appears sound and the Message Passing Interface algorithm that the authors created to enable implementation of the approach to millions of SNPs in tens of thousands of individuals is quite clever. The approach also has many desirable practical features, including allowing for the use of summary statistics for secondary traits rather than requiring individual-level data.

Nevertheless, the simulation results appear to suggest that leveraging cross-trait associations in CTPR only makes a marked improvement in prediction over standard penalized regression when 1) the genetic correlation between traits is large (>0.5 - 0.75 depending on simulation setup), and 2) the percentage of causal SNPs influencing the primary phenotype that also influence the secondary phenotype is also quite large ($>75\%$ according to Figure S1). These situations appear to be the exception rather than the norm in practice. Even complex quantitative phenotypes known to have strong correlations like systolic and diastolic blood pressure report Pearson correlation coefficients of around ~ 0.7 , which is (approximately) the lower bound of when the CTPR approach would appear to have an advantage. Therefore, in most practical situations, it appears that standard published penalized regression procedures like Lasso would provide just as good prediction performance under non-infinitesimal models compared to what is proposed in the manuscript. Moreover, under infinitesimal models, the mixed-model MTGBLUP would appear to be the best choice for prediction.

The approach is applicable only to continuous phenotypes which limits utility. However, the authors mention the approach can be extended to binary disease phenotypes. More detail on this extension would be quite interesting and potentially valuable.

Given the proposed CTPR approach allows for prediction of a primary phenotypes incorporating information from multiple different studies of different secondary phenotypes, why wasn't such an analysis considered for the study of height? That is, why were age of menarche and BMI only considered separately and not jointly for predicting height?

The simulations assumed that all traits were positively correlated with one another but, in the applied example, height and BMI were negatively correlated. Does negative correlation have any impact on the prediction ability of the approach? If a SNP has a positive beta estimate for the primary phenotype but a negative beta estimate for the secondary phenotype (so it increase the primary phenotype but decreases the secondary phenotype), does the prediction algorithm run into problems? I ask because the cross-trait penalty in formula 2 (page 24) is quadratic in nature so it appears the penalty's behavior would differ depending on whether the causal SNP induces positive or negative correlation among the phenotypes. My feeling is that ideally a penalty should be used that is insensitive to differences in sign of the beta coefficients for the primary and secondary phenotypes.

Is there an advantage of using MCP over something like adaptive LASSO that also puts less penalty on

larger coefficients?

Reviewer #2 (Remarks to the Author):

The study introduces new prediction methods and including based on summary statistics. It follows the standard route of exploring method comparisons based on simulation then applying the new methods to real data. I have several major concerns about the conclusions drawn by the authors.

First start with Figure 3. The reader is clearly positioned to believe that MCP & Lasso outperform STGBLUP and MTGBLUP based on simulations with two scenarios that give the impression of covering reality 1M SNPs (3b) showing even more dramatic differences than the small scale simulations (3a). Error bars are small. At first sight, this looks convincing.

Contrast Figure 4 with Figure 3. STGBLUP out performs MCP in each scenario and outperforms Lasso when 1M and 2M SNPs are used in the analysis. Lasso outperforms STGBLUP when 0.15M SNPs are included, but since there is an increased R² for each method as the number of SNPs increases, this implies that the 2M SNPs is the best match for the true genetic architecture.

So Figure 4 clearly shows quite a different result to Figure 3: In Figure 4 STGBLUP way-outperforms Lasso and MCP, but in Figure 3 Lasso and MCP way out-perform STGBLUP. This suggests that the genetic architecture used in the simulations for Figure 3 are not representative of the traits considered in the data application. Since similar genetic architectures are used in Figures 1 and 2, it seems that the authors have chosen simulation scenarios which provide some insight but should be supplementary and should not be the focus of the main paper.

So what underlying genetic architecture could explain the results in Figure 4 where STGBLUP outperforms Lasso and MCP? Supp Figure S5 looks at some alternative simulation scenarios, but those results also don't match the results with the real data. However, Suppl Figure 6 simulates an infinitesimal model, these results are in the supplement because "the infinitesimal genetic architecture (i.e. all SNPs are causal), which is not realistic for most phenotypes". While I agree that it is not realistic that all SNPs are causal, the infinitesimal simulation provides the best fit to the real data, which means that the infinitesimal architecture with real genotype data (LD etc) may be the best approximation to the highly polygenic architecture.

In Figure 4, there is no advantage of MTGBLUP over STGBLUP, given the small genetic correlations in the real data, this is not really surprising. My assumption is that MTGBLUP and Lasso/MCP + CTPR (NHS Stat/Indiv) use the same data. If this is true, then the reader needs explanation why there is an increase in R² compared to Lasso/MCP without CTPR. Figures 1 and 2 show no gain of CTPR when the genetic correlation is 0.1. These results don't add up.

Varying genetic correlations should be added for the infinitesimal model simulations.

Figure 4 shows the comparison between CTPR (NHS Stat and NHS Indiv) - a big increase in R² from using the full genotype data compared to the summary statistics. Figure 2 simulations show no such difference when the genetic correlation between the traits is low. So some explanation of this is needed.

Figure 4, also compares CTPR when using summary statistics data from the largest GWAS, which introduces increased gain based on sample size. The authors do not consider using the summary statistics from GIANT for the primary trait, as this alone gives variance explained in height of about 15%. So this paper is very much positioned to be about methods and not about results.

Lastly, Supp Figure 3 provides a conclusion about the value of using multiple secondary traits, but the simulation scenarios don't represent the real data situation. Would prediction for height be improved by using height, AAM and BMI together in a predictor? Based on real data and simulation under a realistic architecture.

So in conclusion, this paper has made an application of the methods it presents to real data, however the result in itself is not of interest because the variance explained in height $< 5\%$ is much less than the 15% explained by a vanilla predictor from the latest height GWAS. This then is a methods comparison paper which spends much effort on exploring the methods under simulation scenarios, but the simulation scenarios are not realistic of real data, so they provide a misleading assessment.

I suggest that the paper is reworked, first presenting the real data application and then using simulation to find a genetic architecture that best fits the data. Then use that baseline to explore impact of changing correlations, use of > 2 traits etc. The compute time seems to be a major limiting factor for the new proposed method and so the summary statistics version is of more interest. Use of GIANT height & BMI, Reprogen AAM in a summary stats only predictor would be of most interest to readers to see how much variance explained can be increased above the 15% given in the primary paper

Here are more specific comments (some of these were made in the course of reading and will become redundant as specific comments based on reworking suggested about, but in some cases the sentiment of the comment will hold through to the revised version).

The authors should cite other studies that have used the lasso method in genetic risk prediction eg PubMed ID 23203348, 24550740 and genomic selection. Just typing "BLUP LASSO" into Google Scholar brings up a rich uncited literature comparing and combining the methods.

As least in methods, bottom of page 27 and at first mention in main text, and in legend of Figure 1, clarify that definition of heritability is variance explained by the causal SNPs included in the simulation.

For Figures, clarify how they inter-relate eg Figure S1 is the same as Fig 1c when proportion of shared variants = 1.

Top of Page 7, "As expected..", explain that when $r_g=1$, it is equivalent to doubling sample size, and point out that the R^2 at $r_g=0$ in Fig 1b, is the same as the $r_g=1$ in Fig 1a.

In general, I think it helps to provide boundaries. For example, at the end of discussion of Fig S3, point out you have only considered scenarios where r_g between primary and other traits is 0.5 and that of course at the extreme of an r_g or 1 between all traits including n traits is equivalent to increasing sample size n -fold.

A standard expectation is that $r_p = r_g$. In simulations here r_e is set to zero, explore the impact of this assumption when the same data are used for both primary and secondary trait.

Provide in the Methods the equation for the upper bounds. It seems strange that the theoretical multitrait prediction accuracy does not depend on the heritability of the second trait (FigS2 a vs b), and my expectation is found in simulation results "These results showed that larger heritability for secondary traits help PA of the primary trait even if the genetic correlation between traits remains the same."

Add the theoretical maximum to Figure 2 and provide explanation is the R2 is greater than the maximum. Repeat simulation excluding relatives?

Add supplementary figure of only using causal SNPs, and show if R2 agrees with theoretical max

Add to Fig S2, $h^2_1=0.25$

"As shown in Fig 1c.", check logic

Figure 1 Legend. Clarify that heritability of 50% means variance explained by the causal SNPs (& top of page 7). The title has "per-SNP heritability". Clarify that this is changing with the number of causal SNPs. It is not clear whether the training samples provide two phenotypes or 1. I assume they provide 1 and the sample is divided into two. This needs to be clarified also in the text. How many simulation replicates. Does small number of replicates explain the fact that there is variation between methods at genetic correlation of 0.

Page 11 "We simulated two phenotypes and used 14,800 samples for a training set and 5,900 for a validation set". This is ambiguous, did all samples have two phenotypes or do you have two sets with different phenotypes.

Then a few lines later "Similar to previous simulations, we used true SNP coefficients and single SNP summary statistics using simple linear regression based on 7,400 and 14,800 samples, respectively. "The Figure 2 legend doesn't help where the information in brackets in the legend box is incomplete and the Figure text has too much shorthand that I find it ambiguous. Suggest you take a few more words.

Page 11 English "contradicting to the"

Figure 1-3 Make y-axis labels consistent

Figure 4 y-axis R2 for height in UK Biobank data – I assume it is "We considered human height (HGT) as the primary trait and age at menarche (AAM) or body mass index (BMI) as the secondary trait from the NHS cohort data and the UK Biobank data" This is ambiguous is the "The R2 for predicting height in the UK Biobank from training data from ? on height and age at menarche (AAM) or body mass index (BMI)"

Figure 4, Put 0.15M and 1M results in supplement

The introduction discusses prediction of disease risk as an important goal, however this paper focusses on prediction of quantitative traits. The discussion states "Second, the current models described here are only applicable to continuous traits. However, it can be readily extended to binary or case-control data using logistic regression model."

While I agree, in principle the extension to binary traits is easy, and this can likely be demonstrated in simulation, I think the reality of disease data mean that the extension in practice may have hurdles. Maybe use of summary statistics will overcome some issues. I feel it would be prudent to express more caution until tested.

Page 17. Add graphs/tables comparing compute time.

The value of Table 1 over Figure 4 is not clear. Place in supplement. "Slope" as a header is vague. And the lower slope from BLUP is not discussed.

Page 25. If tuning parameters from five-fold cross validation are used then they should also be applied to GBLUP methods to determine the optimum shrinkage parameter.

Is NHS part of GIANT or REPROGEN GWAS summary statistics. If so, does this impact results.

Check impact of removing relatives.

Page 21 seems unnecessary to use just women.

Reviewers' comments:

Reviewer #1 (Remarks to the Author):

This well-written manuscript proposes an approach to improve polygenic risk prediction in GWAS studies of a continuous phenotype by leveraging possible shared genetic effects with secondary phenotype(s) correlated with the primary phenotype. To do this, the authors start with a penalized regression procedure (either Lasso or MCP) for each phenotype and then incorporate an additional cross-trait penalty function using a Laplacian quadratic penalty that is a function of the squared difference in SNP beta coefficients between the primary and secondary phenotype. A novel computational algorithm is then utilized to allow the method to scale to standard GWAS projects. The method is illustrated using both simulated data and a GWAS study of height that leverages additional cross-trait information from independent genetic studies of age of menarche and BMI. The statistical algorithm appears sound and the Message Passing Interface algorithm that the authors created to enable implementation of the approach to millions of SNPs in tens of thousands of individuals is quite clever. The approach also has many desirable practical features, including allowing for the use of summary statistics for secondary traits rather than requiring individual-level data.

We thank the reviewer for the positive remark and summarizing our work concisely.

Nevertheless, the simulation results appear to suggest that leveraging cross-trait associations in CTPR only makes a marked improvement in prediction over standard penalized regression when 1) the genetic correlation between traits is large (>0.5 - 0.75 depending on simulation setup), and 2) the percentage of causal SNPs influencing the primary phenotype that also influence the secondary phenotype is also quite large ($>75\%$ according to Figure S1). These situations appear to be the exception rather than the norm in practice. Even complex quantitative phenotypes known to have strong correlations like systolic and diastolic blood pressure report Pearson correlation coefficients of around ~ 0.7 , which is (approximately) the lower bound of when the CTPR approach would appear to have an advantage. Therefore, in most practical situations, it appears that standard published penalized regression procedures like Lasso would provide just as good prediction performance under non-infinitesimal models compared to what is proposed in the manuscript. Moreover, under infinitesimal models, the mixed-model MTGBLUP would appear to be the best choice for prediction.

We thank the reviewer for noting the improvement and simulation settings of Figures 1 and S1 (i.e. in some settings, substantial improvement at genetic correlation > 0.5 or proportion of shared SNPs > 0.7). We would like to note that this first simulation study was designed to compare the performance of methods when they were at high power (i.e. low uncertainty of the estimated power). Figure 1a clearly depicted the gain in prediction power for CTPR across a range of cross-trait heritability. Starting from setting of Figure 1a as the baseline, we then tried to demonstrate the situation by changing the 3 important parameters (sample size N , total number of SNPs and total number of causal SNPs) one at a time, so that we know how each parameter would affect the relative gain in power by CTPR. For example, Figure 1b vs. Figure 1a shows how increasing sample size would boost power for both but CTPR still gain in power. Because it is already so powerful that single trait methods (Lasso and MCP) achieve an R^2 that is close to the theoretical maximum and therefore the further gain by CTPR appears not so impressive. The same applies to Figure 1c. While in Figure 1d, when the problem becomes harder (more causal SNPs while keeping the same total heritability), the relative gain by CTPR becomes more apparent. The simulation of Figures 1a-1d are to show the effect direction of the 3 parameters (sample size N , total number of SNPs and total number of causal SNPs) in an efficient way.

To assess the gain of power in the norm of practice, we recommend to use GWAS data. Our large-scale GWAS analyses (Figure 4) strongly support the substantial gain of power by CTPR. With 2M SNP data, the relative gains in prediction R^2 using our multi-trait approaches are 69.5% for Lasso+CTPR (118.3% for MCP+CTPR) using AAM and 70.4% (121.6%) using BMI.

The ‘per-SNP heritability’ is one of the key factors to impact the prediction R^2 for Lasso-based methods. In p8 of the manuscript, we noted that “the smaller per-SNP heritability the larger relative gain in prediction accuracy by the multi-trait methods”. In the reality for complex diseases/traits, the heritability explained by any single SNP is extremely smaller, so we expected larger improvement by multi-trait methods. We conducted additional simulation with different number of causal SNPs (i.e. 2,000 or 3,000) randomly chosen from the 2M SNP data in Figure 4. Trait-heritability and genetic correlation among three traits were set the same as the real GWAS datasets. New figure (Figure 5) clearly shows that the gain in prediction accuracy over single-trait methods is getting larger as more causal SNPs (less per-SNP heritability) explain the trait (2M(3000) vs. 2M(2000)). And the prediction accuracy of the real data (Figure 4 and 2M(real) in Figure 5) is similar to that for 3000 simulated causal SNPs (2M(3000) in Figure 5).

To understand the prediction performance with respect to the infinitesimal model, we carried out additional simulation in Supplementary Figure S7 (pasted below). For given sample size and given total trait-heritability, the BLUP based method (Ridge) is always the best for complete infinitesimal scenario. But when the true genetic architecture is deviated from 100% infinitesimal model, the Lasso based method becomes better. In Figure S7, the crossing point of the red (Lasso) and the blue (Ridge/BLUP) lines moves toward 100% causal SNPs as the power to detect causal SNP increases (i.e. by increasing sample size or increasing trait-heritability). For example, for a training sample size of 14,800, the Lasso based method becomes better than BLUP based method even if as much as 70% of all SNPs are causal. As the training sample size of modern GWAS is getting larger and larger, such as the 500,000 UK Biobank sample release is getting ready soon, we expect the Lasso based method will be the better choice for more and more genetic traits.

In summary, we have observed gain of power from multi-trait models as compared to single trait models. The gain of power is more apparent for more difficult prediction problem (e.g. GWAS scale). As sample size getting larger and larger, we will have more power to select the causal SNPs, the Lasso-based multi-trait model is better than BLUP-based multi-trait model in most realistic genetic architecture.

The approach is applicable only to continuous phenotypes which limits utility. However, the authors mention the approach can be extended to binary disease phenotypes. More detail on this extension would be quite interesting and potentially valuable.

We fully agree with the reviewer's comment. Our current tool can be directly applied to binary traits by treating them as continuous traits with values of 1 and 0. The output predicted score might be out of the [0,1] range but a predicted outcome can be assigned by setting a threshold on the predicted score (e.g. output score > 0.5 would predict Yes, and No otherwise). The prediction accuracy, such as the area under the ROC curve (AUC), can be evaluated by varying the threshold. As we mentioned in the paper, as a more elegant approach, it is quite straightforward to extend the current model to binary data using logistic regression. We now are implementing this, so it will be available soon.

Given the proposed CTPR approach allows for prediction of a primary phenotypes incorporating information from multiple different studies of different secondary phenotypes, why wasn't such an analysis considered for the study of height? That is, why were age of menarche and BMI only considered separately and not jointly for predicting height?

We appreciate the reviewer's suggestion. We have considered both AAM and BMI jointly as secondary traits although these were not presented in the original manuscript. More secondary traits lead to better prediction performance in general but the gain will be smaller when genetic correlations among secondary traits are stronger (see Figure S3). In our real GWAS analysis with 2M SNP data, the genetic correlation between AAM and BMI was -0.52 which is much larger than genetic correlations between HGT-AAM (0.18), HGT-BMI (-0.13). Thus, we can expect that three-trait results are quite similar to two-trait results. To prove this, we analyzed three traits jointly using CTPR and MTGBLUP methods and added the results to Table 1. The gain in prediction accuracy of three-trait results using MTGBLUP was much less than the gain using our CTPR methods compared to corresponding two-trait results although the gain using our CTPR methods was also not large. The prediction accuracy of three-trait results improved from $R^2=0.0434$ (AAM), 0.0485 (BMI) to $R^2=0.0504$ (AAM+BMI) using Lasso+CTPR and from $R^2=0.0431$ (AAM), 0.0484 (BMI) to $R^2=0.0503$ (AAM+BMI). Using MTGBLUP, the prediction accuracy of three-trait results changed from $R^2=0.0413$ (AAM), 0.0415 (BMI) to $R^2=0.0415$ (AAM+BMI) in Table 1.

The simulations assumed that all traits were positively correlated with one another but, in the applied example, height and BMI were negatively correlated. Does negative correlation have any impact on the prediction ability of the approach? If a SNP has a positive beta estimate for the primary phenotype but a negative beta estimate for the secondary phenotype (so it increase the primary phenotype but decreases the secondary phenotype), does the prediction algorithm run into problems? I ask because the cross-trait penalty in formula 2 (page 24) is quadratic in nature so it appears the penalty's behavior would differ depending on whether the causal SNP induces positive or negative correlation among the phenotypes. My feeling is that ideally a penalty should be used that is insensitive to differences in sign of the beta coefficients for the primary and secondary phenotypes.

We fully understand the concern raised by the reviewer. This problem is simultaneously resolved by the coefficient rescaling step (p18-p19 in the manuscript). Before performing CTPR, we first compute all marginal beta coefficients for all traits using simple linear regression (or use the beta coefficient from summary statistics), and then re-scale the secondary traits to ensure the slope for beta coefficients of the primary trait regressed on beta coefficients of the secondary traits to be equal to 1. This procedure is designed to ensure all beta estimates for all traits are on the same scale. As an effect, they will become positively correlated after rescaling. For example, since HGT and BMI were negatively correlated, the slope for beta coefficients of HGT regressed on beta coefficients of BMI was negative. We used beta coefficients of BMI (secondary trait) multiplied by the slope for HGT~BMI regression and thus two beta coefficients should be comparable (both scale and sign). We added a sentence in the discussion section of the revised manuscript to clarify this issue.

Is there an advantage of using MCP over something like adaptive LASSO that also puts less penalty on larger coefficients?

We thank for the reviewer to ask the question on the advantage of using MCP over adaptive Lasso. Although the adaptive Lasso adds weights to counteract the known issue of Lasso estimates being biased through

imposing less penalty on large coefficients, the author of MCP paper [Zhang 2010 Annals of statistics] stated that effectiveness of the adaptive Lasso for selection consistency essentially requires the initial estimator to be larger than a (possibly unspecified and random) threshold for most large/non-zero beta estimates and smaller than the same threshold for most small/zero beta estimates (See p927 in MCP paper). We expect that the predictive performance using adaptive Lasso+CTPR might be better than Lasso+CTPR but it remains for future work to evaluate adaptive Lasso+CTPR and comparison with MCP+CTPR.

Reviewer #2 (Remarks to the Author):

The study introduces new prediction methods and including based on summary statistics. It follows the standard route of exploring method comparisons based on simulation then applying the new methods to real data. I have several major concerns about the conclusions drawn by the authors.

First start with Figure 3. The reader is clearly positioned to believe that MCP & Lasso outperform STGBLUP and MTGBLUP based on simulations with two scenarios that give the impression of covering reality 1M SNPs (3b) showing even more dramatic differences than the small scale simulations (3a). Error bars are small. At first sight, this looks convincing.

We thank the reviewer for this positive remark.

Contrast Figure 4 with Figure 3. STGBLUP out performs MCP in each scenario and outperforms Lasso when 1M and 2M SNPs are used in the analysis. Lasso outperforms STGBLUP when 0.15M SNPs are included, but since there is an increased R^2 for each method as the number of SNPs increases, this implies that the 2M SNPs is the best match for the true genetic architecture.

So Figure 4 clearly shows quite a different result to Figure 3: In Figure 4 STGBLUP way-outperforms Lasso and MCP, but in Figure 3 Lasso and MCP way out-perform STGBLUP. This suggests that the genetic architecture used in the simulations for Figure 3 are not representative of the traits considered in the data application. Since similar genetic architectures are used in Figures 1 and 2, it seems that the authors have chosen simulation scenarios which provide some insight but should be supplementary and should not be the focus of the main paper.

So what underlying genetic architecture could explain the results in Figure 4 where STGBLUP outperforms Lasso and MCP? Supp Figure S5 looks at some alternative simulation scenarios, but those results also don't match the results with the real data. However, Suppl Figure 6 simulates an infinitesimal model, these results are in the supplement because "the infinitesimal genetic architecture (i.e. all SNPs are causal), which is not realistic for most phenotypes". While I agree that it is not realistic that all SNPs are causal, the infinitesimal simulation provides the best fit to the real data, which means that the infinitesimal architecture with real genotype data (LD etc) may be the best approximation to the highly polygenic architecture.

We really thank the reviewer for pointing out. The simulation settings in Figures 1-3 were designed such that high prediction power can be achieved, therefore low standard error of power estimation from simulation runs. The low uncertainty of power estimation is important when relating the prediction accuracy with important parameters, including cross-trait heritability, sample size, total number of genotyped SNPs, total number of causal SNPs, individual level data vs. summary statistics, our method vs. most recent cross-trait methods (MTGBLUP). These results are important for the first part of the paper to introduce the new method and examine its behavior under the change of these important parameters.

In the second part of the paper, the new method was applied to real data and performance of different methods were compared. The prediction accuracy of real data for complex trait heavily depends on sample size as the SNP-chip heritability is not close to its total heritability and the prediction R^2 using conventional whole-genome methods or GWAS significant hits is even smaller than the SNP-chip heritability.

To answer the question why Figure 3 and some settings in Figure 4 appear to show different performance comparison between methods. We conducted additional simulations to demonstrate how the power to detect causal SNPs would affect the relative performance between Lasso based method and BLUP based method. We simulated 5,000 SNPs and maintained total heritability at $h^2=0.2, 0.4$ or 0.6 and sample size at $N=3,700, 7,400$ or $14,800$, while we changed number of causal SNPs (i.e. per-SNP heritability). The prediction R^2 were averaged over 100 replications. We confirmed that Lasso results using our CTPR were the same as Lasso using

‘glmnet’ and STGBLUP results were the same as Ridge using ‘glmnet’. Simulation results are presented below (also in Supplementary Figure S7). For given sample size and given total trait heritability, the BLUP based method is always the best for the 100% infinitesimal scenario. But when the true genetic architecture is deviated from 100% infinitesimal model, the Lasso based method becomes better. Importantly, the crossing point of the red (Lasso) and the blue (Ridge/BLUP) lines moves toward 100% causal SNP as the power to detect causal SNP increases (i.e. by increasing sample size or increasing total heritability). For example, for a training sample size of 14,800, the Lasso based method becomes better than BLUP based method even if as much as 70% of all SNPs are causal. As the training sample size of modern GWAS is getting larger and larger, such as the 500,000 UK Biobank sample release is getting ready soon, we expect the Lasso based method will be the better choice for more and more genetic traits.

To summarize, it is not the genetic architecture to make a difference in Figure 3 vs. first few settings in Figure 4. It is the different power to detect causal SNPs to make the difference. As also showed in Figure 4, when combining other traits, essentially increase effective sample size, hence larger power to detect causal SNPs, Lasso based method (CTPR) becomes better than BLUP based method (MTGBLUP).

To address the question whether the infinitesimal simulation provides the best fit to the real data, we conducted additional simulations below (also in Figure 5). It shows for a range of single trait and multi-trait methods, the prediction performance based on model assuming 3000 causal SNPs out of the 2M SNPs matches closely to the real data. This simulation suggests the underlying genetic architecture of height used for analysis in Figure 4 is far from infinitesimal model. As we addressed in below items in more detail, Tables S21 –S23 showed the genetic architecture of a range of complex disease and traits are far from infinitesimal model.

In Figure 4, there is no advantage of MTGBLUP over STGBLUP, given the small genetic correlations in the real data, this is not really surprising. My assumption is that MTGBLUP and Lasso/MCP + CTPR (NHS Stat/Indiv) use the same data. If this is true, then the reader needs explanation why there is an increase in R² compared to Lasso/MCP without CTPR. Figures 1 and 2 show no gain of CTPR when the genetic correlation is 0.1. These results don't add up. Varying genetic correlations should be added for the infinitesimal model simulations.

We thank the reviewer for the very sharp comment. First, yes, the same data were used for CTPR and MTGBLUP. As the reviewer mentioned, it is not surprising that there is a small gain in prediction R² of MTGBLUP over STGLBUP (~0.003 for AAM and ~0.001 for BMI) because genetic correlations between HGT-AAM=0.18 and HGT-BMI=-0.13, which were not high. However, the gain in prediction R² using CTPR over Lasso or MCP is relatively large in Figure 4 (0.215 for AAM and 0.286 for BMI). To obtain more insight into this result, we would refer the reviewer to simulation results in Figures S5a and S5b which show a gain in prediction R² using our multi-trait methods (CTPR) over single-trait methods (Lasso or MCP) becomes larger but a gain using MTGBLUP over STGBLUP is relatively similar when we increase the total number of SNPs and maintain the proportion of causal SNPs at the same level (total number of SNP from 5000 to 10000). It suggests as we genotype more and more SNPs, the relative gain of MTGBLUP vs. STGBLUP remains relatively constant, while the relative gain of CTRP vs. Lasso/MCP is getting larger and larger. The main reason is that our CTPR method contains a cross-trait penalty that has quadratic term of both coefficients, which induce smoothness similar to elastic-net method. Therefore, our CTPR method can identify more causal SNPs than single-trait method (Lasso or MCP) when per-SNP heritability is quite small. Furthermore, since our method utilizes information from other secondary traits, it outperforms GBLUP method.

Figure 4 shows the comparison between CTPR (NHS Stat and NHS Indiv) - a big increase in R² from using the full genotype data compared to the summary statistics. Figure 2 simulations show no such difference when the genetic correlation between the traits is low. So some explanation of this is needed.

We appreciate the reviewer for this incisive remark. In case of large per-SNP heritability (500 causal SNPs explain 50% total heritability) in Figure 2, beta coefficients were much more accurately estimated than those in

real GWAS analyses in Figure 4. Our CTPR method utilizes fixed beta coefficients from summary statistics through the cross-trait penalty term and thus relatively less accurate coefficients from NHS Stat did not produce the comparable predictive performance to full genotype data. If we further examine other prediction results using GIANT Stat or ReproGen Stat in Figure 4, the difference between summary and genotype data were getting smaller. So if more accurate summary statistics with larger sample size are available, this gap would be more narrowed.

Figure 4, also compares CTPR when using summary statistics data from the largest GWAS, which introduces increased gain based on sample size. The authors do not consider using the summary statistics from GIANT for the primary trait, as this alone gives variance explained in height of about 15%. So this paper is very much positioned to be about methods and not about results.

We thank for the reviewer's suggestion. Our paper focused mainly on cross-trait prediction method and provide insight into how shared genetic architecture across traits and diseases could help identify causal variants and improve prediction accuracy. If height is the primary trait of interest, our method can take height summary statistics from other GWAS, such as GIANT, as the secondary trait in the model. But this is not relevant to demonstrate the relative gain by using related but different traits (e.g. BMI and AAM, etc.), so it is not the focus of the paper. We are also working on an extension of the method solely based on summary statistics without the need for individual level genotype even for the primary trait. This will be included in future publication.

Lastly, Supp Figure 3 provides a conclusion about the value of using multiple secondary traits, but the simulation scenarios don't represent the real data situation. Would prediction for height be improved by using height, AAM and BMI together in a predictor? Based on real data and simulation under a realistic architecture.

We thank for the reviewer's remark. As mentioned earlier, we also have considered both AAM and BMI as secondary traits jointly although this was not presented in the original paper. In Figure S3, more secondary traits led to better prediction performance in general but they were helpful only when genetic correlations among secondary traits are small. In our real GWAS analysis with 2M SNP data, genetic correlation between AAM and BMI was -0.52 which is much larger than genetic correlation between HGT-AAM (0.18), HGT-BMI (-0.13). Thus, we can expect that three-trait results are quite similar to two-trait ones. To explain this, we analyzed three traits jointly using CTPR and MTGBLUP methods and added the results to Table 1. The gain in prediction accuracy of three-trait approach using MTGBLUP was much less than the gain using our CTPR methods compared to corresponding two-trait results although the gain using our CTPR methods was also not big. The prediction accuracy of three-trait results improved from $R^2=0.0434$ (AAM), 0.0485 (BMI) to $R^2=0.0504$ (AAM+BMI) using Lasso+CTPR and from $R^2=0.0431$ (AAM), 0.0484 (BMI) to $R^2=0.0503$ (AAM+BMI) using MCP+CTPR. Using MTGBLUP, the prediction accuracy three-trait approach changed from $R^2=0.0413$ (AAM), 0.0415 (BMI) to $R^2=0.0415$ (AAM+BMI).

So in conclusion, this paper has made an application of the methods it presents to real data, however the result in itself is not of interest because the variance explained in height $< 5\%$ is much less than the 15% explained by a vanilla predictor from the latest height GWAS. This then is a methods comparison paper which spends much effort on exploring the methods under simulation scenarios, but the simulation scenarios are not realistic of real data, so they provide a misleading assessment.

I suggest that the paper is reworked, first presenting the real data application and then using simulation to find a genetic architecture that best fits the data. Then use that baseline to explore impact of changing correlations, use of > 2 traits etc. The compute time seems to be a major limiting factor for the new proposed method and so the summary statistics version is of more interest. Use of GIANT height & BMI, Reprogen AAM in a summary stats only predictor would be of most interest to readers to see how much variance explained can be increased above the 15% given in the primary paper

We really thank for the reviewer's suggestions. We would like to note that the comparison of prediction R^2 in this paper (~5%) with the vanilla predictor from GIANT (~15%) is not a fair comparison, as the sample size used in this paper (Figure 4) is 11,473 while in GIANT, it is 339,224, more than 30 times larger sample size. If we could apply our method on the GIANT samples, we would expect a larger prediction R^2 than 15%. We would also like to note that we have examined an extensive range of simulation scenarios to evaluate the method and compare methods in real large scale GWAS data using real traits. These results consistently suggest that the CTPR method works better than single trait method and other existing multi-trait method. The reviewer has raised question about single trait Lasso based method and BLUP based methods, we have provided additional simulations in the above response to illustrate the phenomenon.

We performed additional simulation studies to find a genetic architecture that best fits the real data and then to explore impact of changing per-SNP heritability using that baseline with 2M SNP data. We mimicked real large-scale GWAS datasets through considering the same trait-heritability and the same genetic correlation among three traits using the NHS cohort data and the UK Biobank data. Two different number of causal SNPs (i.e. 2,000 and 3,000) were randomly chosen from 2M SNPs and used to evaluate effect of per-SNP heritability on prediction accuracy. We set the trait-heritability of three traits to 40.4% (Sim HGT), 25.3% (Sim AAM) and 20.0% (Sim BMI) and the genetic correlation between Sim HGT and Sim AAM to 0.184, that between Sim HGT and Sim BMI to -0.133 and that between Sim AAM and Sim BMI to -0.527. We considered Sim HGT as the primary trait and Sim AAM or Sim BMI as the secondary trait. Prediction R^2 were computed using GBLUP (STGBLUP, MTGBLUP) and the proposed prediction methods (MCP/MCP+CTPR, Lasso/Lasso+CTPR). The results with 3,000 causal SNPs in the simulation are quite similar to (slightly larger than) corresponding real data analyses. The overall prediction R^2 increases and our CTPR performed better than GBLUP as less causal SNPs (larger per-SNP heritability) are considered in the simulation below (Figure 5). These results explain the difference between Figure 3 and Figure 4. With large per-SNP heritability (500 SNPs) in Figure 3, our CTPR methods including single-trait approaches performed much better than GBLUP methods but our single-trait methods performed slightly worse than GBLUP methods with small per-SNP heritability (>3000 SNPs) in Figure 5, which is similar to the real data results in Figure 4. This is also consistent with our additional simulation in Figure S7, explained above.

To further investigate the true genetic architecture for real traits and disease, we estimated the proportion of causal SNPs for human height using two-component normal mixture models [McLachlan et al. 2006 Bioinformatics]. The proportion of causal SNPs were estimated as ~1.39% with summary statistics from NHS cohort (N=11,473), ~5.09% from UK Biobank (N=113,851) and ~9.14% from GIANT consortium (N=253,288) (Tables S21, S22). Also, the proportion of causal SNPs using above simulated data with 3,000 causal SNPs was estimated as ~1.41%, which explains why prediction R^2 with those data were similar to our real data results with 2M SNPs. Furthermore, the estimated proportions of causal SNPs for many other clinical and disease phenotypes (e.g. asthma, hypertension, type 2 diabetes and several cancer types) are less than those for human height with the same sample size (Tables S22, S23). Since the genetic architecture underlying these broad range of phenotypes have a smaller proportion of causal SNPs than human height, we can expect that our multi-trait methods outperform MTGBLUP methods for many phenotypes of interest.

Minor comments

Here are more specific comments (some of these were made in the course of reading and will become redundant as specific comments based on reworking suggested about, but in some cases the sentiment of the comment will hold through to the revised version).

The authors should cite other studies that have used the lasso method in genetic risk prediction eg PubMed ID 23203348, 24550740 and genomic selection. Just typing “BLUP LASSO” into Google Scholar brings up a rich uncited literature comparing and combining the methods.

We thank the reviewer for pointing out these relevant references. We cited and discussed them in the introduction section of the revised manuscript.

As least in methods, bottom of page 27 and at first mention in main text, and in legend of Figure 1, clarify that definition of heritability is variance explained by the causal SNPs included in the simulation.

In the simulation, beta coefficients for multiple traits in the simulation were generated from multivariate normal distribution only for causal SNPs and then other beta coefficients were set to zero. The covariance matrix for multivariate normal distribution were designed with number of causal SNPs, desired heritability and genetic correlation among traits. Therefore, heritability in the simulation becomes variance explained by the causal SNPs as the reviewer stated. We have added additional clarification to method section of the revision as the reviewer suggested.

For Figures, clarify how they inter-relate eg Figure S1 is the same as Fig 1c when proportion of shared variants = 1.

We considered the scenarios with 14,800 training samples (7,400 samples for both traits), 5,000 SNPs, 500 causal SNPs and 50% trait-heritability for both Figure 1c and S1. For Figure 1c, we varied genetic correlation from 0 to 1 while all causal SNPs were shared, but, for Figure S1, we fixed genetic correlation at 0.1, 0.5, 0.7 and 0.9 level while the proportion of shared SNPs were varied from 0 to 1.

Top of Page 7, “As expected..”, explain that when $r_g=1$, it is equivalent to doubling sample size, and point out that the R^2 at $r_g=0$ in Fig 1b, is the same as the $r_g=1$ in Fig 1a.

We thank the reviewer for the very sharp comment. Yes, when genetic correlation=1 and separate samples for two traits are used in simulation, the prediction accuracy for two-trait approach is very similar to those for single-trait approach with double samples. We added one sentence on page 7 to make them clearer.

In general, I think it helps to provide boundaries. For example, at the end of discussion of Fig S3, point out you have only considered scenarios where r_g between primary and other traits is 0.5 and that of course at the

extreme of an r_g or 1 between all traits including n traits is equivalent to increasing sample size n -fold.

Yes, when genetic correlation is close to 1 and different samples for all traits are used, prediction accuracy using n -trait approach would be very similar to that using single-trait approach with n times more samples.

A standard expectation is that $r_p = r_g$. In simulations here r_e is set to zero, explore the impact of this assumption when the same data are used for both primary and secondary trait.

As long as G is independent to E , r_e should have no impact on the effect of r_g . We did not consider $G \times E$ interaction in this method, which could be of potential interest in the extension of the method.

Provide in the Methods the equation for the upper bounds. It seems strange that the theoretical multitrait prediction accuracy does not depend on the heritability of the second trait (FigS2 a vs b), and my expectation is found in simulation results “These results showed that larger heritability for secondary traits help PA of the primary trait even if the genetic correlation between traits remains the same.”

We appreciate the reviewer’s suggestion. We added the following explanation for the theoretical upper bound of prediction accuracy to the method section. We also note that the theoretical upper bounds of prediction accuracy shown in Figures S2a vs S2b are for single trait prediction model with two different sample sizes (i.e. $N=7,400$ or $14,800$).

The theoretical upper bound of prediction accuracy has been suggested based on the principles of population genetics [Yang et al. 2010 Twin Research and Human Genetics]. It can be achieved when we estimate effects in one population and then predict genetic risk in another samples in the same population. The prediction accuracy is defined as the squares of correlation between true phenotype values and predicted ones. The expected value of prediction accuracy is as $E[R^2(y, \hat{y})] \approx \frac{h^2 \text{var}(g)}{[\text{var}(g) + c \cdot \frac{\text{var}(y)}{N}]} = h^2 / [1 + c/h^2 N]$ where g is the additive genetic values, c is the number of causal SNPs, h^2 is the observed heritability and N is the total number of individuals.

Add the theoretical maximum to Figure 2 and provide explanation is the R^2 is greater than the maximum. Repeat simulation excluding relatives? Add supplementary figure of only using causal SNPs, and show if R^2 agrees with theoretical max

Two grey lines in Figure 1 represent theoretical upper bound of prediction accuracy for single-trait (lower) and multi-trait approaches (upper, i.e. double sample size). If the reviewer compares upper grey lines with Lasso+CTPR/MCP+CTPR and lower grey lines with Lasso/MCP, most prediction accuracies using CTPR are less than the theoretical upper bounds. Only in Figures S2b, S2d, we found prediction R^2 using CTPR are slightly larger than the theoretical maximum. This can be explained by the fact that those theoretical maximum were computed based on SNP effects estimated from ordinary least squares method, not penalized least squares method [Yang et al. 2010 Twin Research and Human Genetics]. Due to the non-infinitesimal model (only 10% SNPs were causal) in Figures S2b, S2d, it may be possible that our penalized regression results exceeds the theoretical maximum designed for ordinary regression.

Add to Fig S2, $h_{21}=0.25$

Yes, in Figure S2, $h_2=(0.5,0.25)$, $h_2=(0.5,0.75)$, $h_2=(0.25,0.25)$ and $h_2=(0.75,0.75)$ means $(h_{21}=0.5, h_{22}=0.25)$, $(h_{21}=0.5, h_{22}=0.75)$, $(h_{21}=0.25, h_{22}=0.25)$ and $(h_{21}=0.75, h_{22}=0.75)$.

“As shown in Fig 1c.”, check logic

We modified the sentence as “As shown in Fig. 1c, PA increases as less non-causal SNPs are included.”

Figure 1 Legend. Clarify that heritability of 50% means variance explained by the causal SNPs (& top of page 7). The title has “per-SNP heritability”. Clarify that this is changing with the number of causal SNPs. It is not clear whether the training samples provide two phenotypes or 1. I assume they provide 1 and the sample is divided into two. This needs to be clarified also in the text. How many simulation replicates. Does small number of replicates explain the fact that there is variation between methods at genetic correlation of 0.

Yes, heritability in the simulation means variance explained by the causal SNPs. Thus, per-SNP heritability is changing with the number of causal SNPs while trait-heritability is the same. We averaged over 100 replications and provided standard error for all simulations in the manuscripts.

Page 11 “We simulated two phenotypes and used 14,800 samples for a training set and 5,900 for a validation set”. This is ambiguous, did all samples have two phenotypes or do you have two sets with different phenotypes.

We feel sorry for the confusion. We simulated two phenotypes for all 14,800 training samples and 5,900 validation samples. And then half of training samples (i.e. 7,400) were used for primary phenotype and other half were used for secondary phenotype. That is, only one phenotype was used for each sample in the analysis while two phenotypes were related. We have included additional clarification on page 11 to make them clearer.

Then a few lines later “Similar to previous simulations, we used true SNP coefficients and single SNP summary statistics using simple linear regression based on 7,400 and 14,800 samples, respectively. “The Figure 2 legend doesn’t help where the information in brackets in the legend box is incomplete and the Figure text has too much shorthand that I find it ambiguous. Suggest you take a few more words.

Since we simulated beta coefficients and phenotypic values, we had true beta coefficients (from the joint model with all causal SNPs included simultaneously). Before performing CTPR, we ran simple linear regression for each SNP (phenotype ~ a single SNP to mimic GWAS) using 7,400 and 14800 samples and save all beta coefficients. Then we conducted CTPR with three inputs, true beta coefficients, summary statistics based on 7,400 and 14,800 samples.

Page 11 English “contradicting to the”

We have revised this as “in contradiction to ...”

Figure 1-3 Make y-axis labels consistent

We have removed (%) from Figure 3 to make y axis labels consistent.

Figure 4 y-axis R2 for height in UK Biobank data – I assume it is

“We considered human height (HGT) as the primary trait and age at menarche (AAM) or body mass index (BMI) as the secondary trait from the NHS cohort data and the UK Biobank data” This is ambiguous is the “The R2 for predicting height in the UK Biobank from training data from ? on height and age at menarche (AAM) or body mass index (BMI)”

We feel sorry for the confusion. Both UK Biobank and NHS cohort data contain three phenotypes such as AAM, BMI and height, and NHS cohort data were used for training and UK Biobank for testing. We have clarified this sentence in the revised manuscript as follows. “We considered human height (HGT) as the primary trait and age at menarche (AAM) or body mass index (BMI) as the secondary trait. NHS cohort data were used as a training set and UK Biobank data were used as a validation set.”

Figure 4, Put 0.15M and 1M results in supplement

We thank for the reviewer's suggestion. In order to evaluate the effect of the total number of SNPs included, Figure 4 contains all results with 0.15M, 1M and 2M SNPs. We still feel it is important to show the trend with increasing number of SNPs. If it would be better to move 0.15M, 1M results to supplementary figures, we could do it in the next revision.

The introduction discusses prediction of disease risk as an important goal, however this paper focusses on prediction of quantitative traits. The discussion states "Second, the current models described here are only applicable to continuous traits. However, it can be readily extended to binary or case-control data using logistic regression model."

While I agree, in principle the extension to binary traits is easy, and this can likely be demonstrated in simulation, I think the reality of disease data mean that the extension in practice may have hurdles. Maybe use of summary statistics will overcome some issues. I feel it would be prudent to express more caution until tested.

We fully agree with the reviewer's comment. Although in principle, we still can apply current program to analyze binary disease traits by treating them as 0, 1 continuous trait, and we are now implementing the logistic regression model, which will be available quite soon. We agree to the reviewer that we want to be more cautious until the method is tested on binary traits. We have revised the wording as: "In principle, the current methods can be directly applied to binary traits by treating them as continuous traits with values of 0 and 1 although the performance on binary traits still need to be evaluated using simulation and real GWAS data. As a more elegant approach, it is quite straightforward to extend the current model to binary data using logistic regression, which is now being implemented and will be available soon."

Page 17. Add graphs/tables comparing compute time.

The proposed methods need more time to estimate all SNP effects than GBLUP methods in general but novel MPI algorithm were implemented using parallel computing to reduce computation time and utilize computing resource effectively. Thus, if q computing nodes are used for MPI version of CTPR, it will take q times faster than normal Lasso-based packages. The comparison might not make sense as one method use only one CPU while the other could use multiple CPUs.

The value of Table 1 over Figure 4 is not clear. Place in supplement. "Slope" as a header is vague. And the lower slope from BLUP is not discussed.

The slopes were computed through regressing the phenotypic values on the predicted values. A regression slope larger than 1 implies that difference between the true phenotypic values in a pair of samples is larger than difference between the predicted values. Overall, regression slopes from CTPR are larger than GBLUP due to biased estimates from shrinkage of the non-zero coefficients in CTPR.

Page 25. If tuning parameters from five-fold cross validation are used then they should also be applied to GBLUP methods to determine the optimum shrinkage parameter.

We appreciate the reviewer's suggestion. The multi-trait GBLUP method [Robert et al. 2015 AJHG] we compared with did not have the option to tune any parameters, so we cannot do it in this paper. Habier et al. 2007 has showed that genomic BLUP (GBLUP) method is equivalent to ridge regression BLUP (RR-BLUP) method. In the Robert et al. 2015 AJHG, the authors cited the bivariate ridge regression by Li et al. 2014 Hum. Genet., but did not do a comparison with it. The Li et al. 2014 proposed to use tuning parameters while Robert et al. 2015 did not proposed to tune any parameters. We agree it might be interesting to see how Li et al 2014 compared with Robert et al. 2015 and if tuning any particular parameters in Robert et al. 2015 would give any marginal improvement over existing implementation. This could be done in future work.

Habier D, Fernando RL, Dekkers JCM (2007) The impact of genetic relationship information on genome-assisted breeding values. *Genetics* 177:2389–2397

Is NHS part of GIANT or REPROGEN GWAS summary statistics. If so, does this impact results.

NHS cohort data were used only as a training set, not a validation set. UK Biobank data were used as a validation set but it is not part of GIANT or REPROGEN consortium. We know some NHS samples contribute to GIANT consortium but it should not have impact on the validity of the prediction R^2 because NHS samples were only used as a training set.

Check impact of removing relatives.

We removed individuals that are duplicates (or identical twins), not related individuals (full sibs, half sibs) in each dataset as mentioned in the manuscript.

Page 21 seems unnecessary to use just women.

Because age at menarche is one of secondary phenotypes of interest, we restricted to only women for our real data analysis.

Reviewers' comments:

Reviewer #1 (Remarks to the Author):

This revision clarifies the technical questions I had with CTPR. I continue to think the method is quite innovative. However, the issues that were raised with the evaluation of CTPR and its competitors using simulations still persist somewhat. I wish the authors had taken Reviewer 2's advice and restructured the paper to begin with the applied results first and then create simulated datasets based on the initial structure of the applied studies but then varying different features among reasonable values that would be expected to impact performance (heritability of each trait, cross-trait correlation, per-SNP heritability, proportion of shared SNPs etc). Many of the simulations (either in main text or supplement) assume cross-trait correlations and values of trait heritability that appear (at best) loosely connected to the observations in the real data

I wish there was more exploration of the effect of proportion of shared causal SNPs on PA. The paper only explores this relationship in one simulation setting (Figure S1) and that setting appears a bit disconnected from the applied example (5000 total SNPs with 500 causal; heritability of primary and secondary trait equal to 50%, which is larger than those reported for HGT/AAM/BMI). All remaining simulations assume the proportion of shared causal SNPs is 1, which seems unlikely in reality.

I would have liked to see a version of Figure S7 that assumed the same number of SNPs in the applied study (2 million rather than the 5000 in the simulation setup) and explored the effect of sample size on traits with heritability of either 0.2 or 0.4 (rather than 0.6 shown in the right side of the panel).

The authors report in this revision that the genetic correlation between AAM and BMI is much larger than genetic correlation of AAM-HGT and BMI-HGT. Given this larger correlation, why didn't the authors explore an analysis that considered BMI as the primary phenotype and AAM as the secondary phenotype (or vice versa)? My instinct such an analysis could be a valuable asset to the paper.

The authors estimated the proportion of causal SNPs in a dataset using a two-component mixture model proposed by McLachlan et al. that is tailored for gene-expression analyses rather than GWAS projects. While it might not make much difference, it would be useful for the authors to redo these calculations using a method specifically designed for GWAS projects, like the AVENGEME method of Palla and Dudbridge (AJHG 97: 250-259).

Perhaps I missed it, but I did not see any details regarding computational run times for the CTPR approach or its competitors.

Reviewer #2 (Remarks to the Author):

The concern I raised in my first review was that the simulation results presented in Figure 3 provided quite a different view of the comparison between methods than the results presented in Figure 4 from real data

The authors argue that the simulations provide baseline comparisons varying key parameters. I can understand this argument, testing under scenarios which are computationally efficient and stretches the model to show differences. But in Figure 3 the authors use Prediction R2 computed based on 14,800 training samples, 500 causal SNPs a) 5,000 SNPs; causal = 10% b) 1 Million SNPs 0.05% causal. The results are described as "The overall predictive performance of our methods in small-scale

simulation was uniformly better than that of STGBLUP and MTGBLUP methods (Fig. 3a).” From Figure S7b it is clear that the choice of architecture is at the extreme of the differences between methods and that from Figure 4 and Figure 5, that such an architecture is not representative of real data. In particular Figure 3b described as “The large-scale simulation” makes readers think it is more representative of real data when it is not.

The authors now add simulations (Figure 5) to try and find an architecture that fits real data (figure 4). Even these simulations are not taken to a point that matches the real results. These simulation should be extended (ie > 3000 causal variants out of 2M) to find a better match to the real data.

I agree that both sample size and genetic architecture contribute to the differences between methods and that as sample sizes increase there may well be no difference between Lasso and GBLUP under an infinitesimal model and Lasso may become the method of choice.

My point stands that I feel that the results as presented are not balanced.

My suggestions are:

- 1) Keep Fig 3a as it matches to Fig 1 and 2.
- 2) Replace Fig 3b by one where but the number of SNPs and the % of causal SNPs are the same as in Fig 3a
- 3) Add Fig 7b as Fig 3c Explain where Fig 3a/b fall on this eg by arrows
- 4) Extend scenarios in Figure 5 to include more causal SNPs to get a better fit to the data. 5) Use from 4 architecture to add a further panel in Figure 3.

Most likely these suggestions need to be tweaked to fit the flow, but the sentiment is to provide simulations that are representative of data as this positions the reader best to evaluate the methods.

Response to Reviewers,

We thank the reviewers for their careful critique of our manuscript. The comments were very helpful, and allowed us to conduct a complete revision of the analyses to improve the quality of the manuscript. In this revision, to evaluate the predictive performance of our CTPR using biobank-scale GWAS data, we fully re-conducted the analyses and revised the manuscript as follows.

First, to assess the predictive validity of the CTPR with biobank-scale individual-level GWAS data, the training sample size for both simulation and real data analyses were increased up to $N=437K$.

Second, as requested by the reviewers, the sample size, number of SNPs and genetic architecture for simulation studies matched to those for real data analysis. We first conducted real data analyses using CTPR to obtain baseline information on the genetic structure of GWAS data and extended the baseline model for further simulations. We varied three important parameters (i.e. cross-trait correlation, total number of traits and sample size) one at a time to provide a clear picture on how each parameter would affect the predictive performance of CTPR. All previous small-scale simulation results were moved to supplementary documents to demonstrate the impact from other considerably important parameters such as proportion of shared causal SNPs, number of causal SNPs, trait-heritability and sample size of summary statistics for secondary traits.

Third, we further compared our CTPR with recently published summary statistics-based prediction methods, MTAG (Turley et al. 2018 Nat. Genet) and LDpred (Vilhjálmsón et al. 2015 AJHG) for both simulation and real data analyses because these methods have been widely adopted for genetic risk prediction in practice due to computational feasibility and easy accessibility GWAS summary statistics. As shown, our CTPR outperformed summary statistics-based methods mainly because CTPR fits all SNPs simultaneously using penalized regression while MTAG or LDpred utilizes only marginal SNP effects for all traits.

Fourth, we investigated predictive performance in different populations. The trained coefficients in UK Biobank were tested both in NHS/HPFS/PHS cohort and in a separated set of UK Biobank samples. In general, prediction accuracy (PA) evaluated in NHS/HPFS/PHS was smaller than PA evaluated in UK Biobank likely due to genetic heterogeneity between UK Biobank and NHS/HPFS/PHS cohort.

For your convenience, we provide a point by-point reply to all reviewer's comments with our responses highlighted in blue below.

Reviewer #1 (Remarks to the Author):

This revision clarifies the technical questions I had with CTPR. I continue to think the method is quite innovative.

We thank the reviewer for the positive remark and considering our method innovative.

However, the issues that were raised with the evaluation of CTPR and its competitors using simulations still persist somewhat. I wish the authors had taken Reviewer 2's advice and restructured the paper to begin with the applied results first and then create simulated datasets based on the initial structure of the applied studies but then varying different features among reasonable values that would be expected to impact performance (heritability of each trait, cross-trait correlation, per-SNP heritability, proportion of shared SNPs etc). Many of the simulations (either in main text or supplement) assume cross-trait correlations and values of trait heritability that appear (at best) loosely connected to the observations in the real data.

We appreciate the reviewer's suggestion. As we explained above, we now started with CTPR using up to 437K UK Biobank samples to find the baseline genetic structure of the GWAS data for height and BMI, and set the baseline model for simulation. We then changed three crucial parameters: cross-trait correlation, total number of traits and sample size, one at a time to obtain practical insight on how to increase PA for the primary trait of interest through collecting appropriate secondary traits and more samples. All previous small-scale simulation results were moved to supplementary documents to demonstrate the impact from other less important parameters.

I wish there was more exploration of the effect of proportion of shared causal SNPs on PA. The paper only explores this relationship in one simulation setting (Figure S1) and that setting appears a bit disconnected from the applied example (5000 total SNPs with 500 causal; heritability of primary and secondary trait equal to 50%, which is larger than those reported for HGT/AAM/BMI). All remaining simulations assume the proportion of shared causal SNPs is 1, which seems unlikely in reality.

We fully understand the concern raised by the reviewer. As requested, we have conducted additional simulations for proportion of shared causal SNPs using large-scale UK biobank data that was also utilized in real data analyses. We first chose sample size ($N=30K$), number of trait ($T=2$), cross-trait correlation ($\rho=0.75$) and proportion of shared SNPs ($PS=1$), and then reduced the proportion from 1 to 0.25 (i.e. $PS=1, 0.75, 0.5, 0.25$) to examine how the proportion affects the PA (left panel of the figure below). Here, the cross-trait correlation (ρ) represents the overall correlation of all causal SNPs between the two traits (genetic correlation evaluated at the union of causal SNP sets of the two traits). The PA decreased as the proportion of shared SNPs decreased. We also presented simulation results for various cross-trait correlation with PS fixed at 1 (right panel of the figure below) to compare with new simulations on proportion of shared causal SNPs. Both results showed quite similar trend.

I would have liked to see a version of Figure S7 that assumed the same number of SNPs in the applied study (2 million rather than the 5000 in the simulation setup) and explored the effect of sample size on traits with heritability of either 0.2 or 0.4 (rather than 0.6 shown in the right side of the panel).

In order to proceed with the reviewer's request, we explored the effect of sample size on PA with P=1M SNPs (we did not use 2M SNPs because 1M SNPs are more computationally tractable for many combinations of sample sizes and heritabilities but still have the same magnitude of SNP numbers the reviewer would like to explore) and 25% or 45% of heritability in Figure 2c, which were the similar genetic structure of the real data for BMI and height. Three different sample sizes were considered such as N=30K, 200K and 437K samples. The simulation results showed that the sample size have huge impact on PA and including more samples helps to improve the PA in general.

The authors report in this revision that the genetic correlation between AAM and BMI is much larger than genetic correlation of AAM-HGT and BMI-HGT. Given this larger correlation, why didn't the authors explore an analysis that considered BMI as the primary phenotype and AAM as the secondary phenotype (or vice versa)? My instinct such an analysis could be a valuable asset to the paper.

We really thank for the reviewer's suggestion. To investigate the reviewer's comment, we have conducted CTPR using AAM and BMI as only traits of interest. The prediction results are given below. For this analysis, samples were restricted to only women (N=176K) due to AAM, and used common SNPs between UK Biobank and NHS/HPFS/PHS cohort because UK Biobank was used as a training set and NHS/HPFS/PHS cohort was used as an independent validation set. There is a clear gain in PA using our cross-trait analysis and PA for AAM-BMI (8.13% using Lasso+CTPR) is a bit larger than PA for BMI-AAM (7.21% using Lasso+CTPR). However, in the paper, we would like to utilize full samples in UK Biobank (N=~457K) to increase PA as much as possible and heritability of height ($h^2 \sim 45\%$) is much larger than AAM ($h^2 \sim 25\%$) and BMI ($h^2 \sim 22\%$) and thus height was used as the only primary trait of interest in the manuscript. In addition, we included other secondary phenotypes, such as hip circumference, waist circumference and waist-hip ratio. The prediction R^2 for those traits were presented in the manuscript. We observed the genetic correlation between height-hip circumference ($\rho = -0.30$) is a little bit larger than that between height-BMI ($\rho = -0.13$) but the PA of height-hip circumference is similar to that of height-BMI likely because heritability of BMI is larger than that of hip circumference.

Primary	Second	Method	R^2	MSE	Slope	# Nzbeta
AAM	x	Lasso	0.0742	3.9825	0.4595	4712
	x	MCP	0.0678	4.6423	0.4245	1751
AAM	BMI	Lasso+CTPR	0.0813	3.8943	0.4956	12310
		MCP+CTPR	0.0762	5.0044	0.4281	3808
BMI	x	Lasso	0.0658	89.7593	0.5130	5546
	x	MCP	0.0598	91.1167	0.3981	3130
BMI	AAM	Lasso+CTPR	0.0721	94.0888	0.4975	17298
		MCP+CTPR	0.0672	115.3301	0.4289	4816

The authors estimated the proportion of causal SNPs in a dataset using a two-component mixture model proposed by McLachlan et al. that is tailored for gene-expression analyses rather than GWAS projects. While it might not make much difference, it would be useful for the authors to redo these calculations using a method specifically designed for GWAS projects, like the AVENGEME method of Palla and Dudbridge (AJHG 97: 250-259).

Thanks for reviewer's suggestion. Previously, we estimated the proportion of causal SNPs using a two-component normal mixture model to investigate non-infinitesimal genetic architecture of various phenotypes. As the reviewer suggested, the AVEMGEME method (Palla et al. 2015 AJHG) provides a fast way to estimate the proportion of causal SNPs as well as the variance explained by SNPs using multiple polygenic score tests based on markers with p-values in different intervals. Recently, a newer method for GWAS data, GENESIS (Zhang et al. 2018 Nat. Genet) was proposed to estimate the proportion of underlying susceptibility SNPs

utilizing two or three-component normal mixture model. The model of GENESIS is quite similar to the model we tried while it seems to generate more conservative results in general. For example, we estimated 5.09% for height and 2.18% for BMI using UK Biobank (N=113K) as causal but the GENESIS method estimated 1.48% for height (N=253K) and 1.40% for BMI (N=124K) as susceptibility SNPs. The main reason we estimated the proportion of causal SNPs in the manuscript was to show many complex traits have non-infinitesimal genetic architecture and they generally have smaller proportion of causal SNPs than height. New estimation from the GENESIS also support this observation and thus we revised the manuscript based on the new results from the GENESIS instead.

Perhaps I missed it, but I did not see any details regarding computational run times for the CPTR approach or its competitors.

We appreciate the reviewer's question on computation time of CTPR. We estimated the computation time and memory usage using a large-scale simulation as follows. To assess the computational feasibility of CTPR for biobank-based GWAS data, we simulated data using N=437K individuals and P=1M SNPs from UK Biobank, which requires ~1.7TB of memory with float data type (i.e. $437K * 1M * 4B \approx 1.7TB$). The CTPR run on 40 cores (Intel Xeon CPU 2.1 GHz) with 48GB of memory for each core, total of ~1.9TB of memory, for up to 7 days to complete the analyses with 40 core-groups (which generates exact solution because all coefficients are updated sequentially in core-group). The running time of CTPR depends linearly not only on the sample size (N) and the number of SNPs (P) but also on the number of core-group (q), which represents $O(NPq)$. With 10 core-groups (which generates approximate solution because coefficients within 4 cores are updated simultaneously), the running time of CTPR dropped to ~1.75 days and it still generated almost the same predictive performance as exact solution due to good convergence. Even when sample size increases, the running time is able to remain similar because larger sample size increases likelihood of convergence and therefore less number of core-groups are needed. Furthermore, we can now utilize public cloud platforms such as Amazon web services (AWS), Google cloud platform (GCP) and Microsoft Azure for CTPR to execute on large-sample GWAS analyses in a more efficient and time-saving manner.

Reviewer #2 (Remarks to the Author):

The concern I raised in my first review was that the simulation results presented in Figure 3 provided quite a different view of the comparison between methods than the results presented in Figure 4 from real data.

We thank for the reviewer's clarification on the concern in the first review. We now carried out new simulations with much larger sample sizes and number of SNPs such that they represent settings similar to real data. More detailed explanations are given below.

The authors argue that the simulations provide baseline comparisons varying key parameters. I can understand this argument, testing under scenarios which are computationally efficient and stretches the model to show differences. But in Figure 3 the authors use Prediction R2 computed based on 14,800 training samples, 500 causal SNPs a) 5,000 SNPs; causal =10% b) 1 Million SNPs 0.05% causal. The results are described as "The overall predictive performance of our methods in small-scale simulation was uniformly better than that of STGBLUP and MTGBLUP methods (Fig. 3a)." From Figure S7b it is clear that the choice of architecture is at the extreme of the differences between methods and that from Figure 4 and Figure 5, that such an architecture is not representative of real data. In particular Figure 3b described as "The large-scale simulation" makes readers think it is more representative of real data when it is not.

We fully understand the reviewer's concern. In the new manuscript, we first carefully investigated the genetic structure of the real data and then designed the baseline model for simulation. The three crucial parameters (i.e. cross-trait correlation, total number of traits and sample size) were varied to obtain information on how each

parameter have impact on PA of the primary trait of interest. New simulation results in Figure 2 showed that our CTPR outperformed both summary statistics-based methods (e.g. MTAG, LDpred) and GBLUP-based method for large number of samples (N=437K) and large number of SNPs (P=1M) representing similar settings in real GWAS datasets.

The authors now add simulations (Figure 5) to try and find an architecture that fits real data (figure 4). Even these simulations are not taken to a point that matches the real results. These simulation should be extended (ie > 3000 causal variants out of 2M) to find a better match to the real data.

I agree that both sample size and genetic architecture contribute to the differences between methods and that as sample sizes increase there may well be no difference between Lasso and GBLUP under an infinitesimal model and Lasso may become the method of choice.

We appreciate the reviewer's incisive comments. In the new manuscript, we found a genetic architecture that fits real data (P=1M SNPs, N=30K, 200K, 437K samples) for height, which have 6,800-12,600 (depending on the sample size) causal SNPs explaining 45% of phenotypic variance. Using this genetic architecture as the baseline model, we showed CTPR performed better than GBLUP-based method as well as summary statistics-based methods (MTAG, LDpred) with various parameter values. The proportion of causal SNPs for height is expected to be 1.48% of SNPs (Zhang et al. 2018 Nat. Genet) and those for other clinical phenotypes and disease traits are expected to have a smaller proportion than height. Because the genetic architecture underlying these broad ranges of phenotypes have proportion of causal SNPs far from the infinitesimal model, we expect that CTPR will perform well for many phenotypes of interest. Furthermore, we found Lasso-based methods perform increasingly better than Ridge/GBLUP based methods as training sample size increases when the same proportion of causal variants were fixed in our simulation studies. As the training sample size of modern GWAS is getting larger and larger (e.g. >500K), we expect our penalized regression based multi-trait approach will be the better choice for even more genetic traits in the future.

My point stands that I feel that the results as presented are not balanced.

My suggestions are:

- 1) Keep Fig 3a as it matches to Fig 1 and 2.
- 2) Replace Fig 3b by one where but the number of SNPs and the % of causal SNPs are the same as in Fig 3a
- 3) Add Fig 7b as Fig 3c Explain where Fig 3a/b fall on this eg by arrows
- 4) Extend scenarios in Figure 5 to include more causal SNPs to get a better fit to the data.
- 5) Use from 4 architecture to add a further panel in Figure 3.

Most likely these suggestions need to be tweaked to fit the flow, but the sentiment is to provide simulations that are representative of data as this positions the reader best to evaluate the methods.

Thanks for the reviewer's detailed suggestions. We have revisited the manuscript with the reviewer's suggestions in mind and then fully revised the manuscript with new simulation and real data results. We added a new figure on overview of CTPR method (new Figure 1) and replaced the previous small-scale simulation by new large-scale simulation that matched to the real data analysis (new Figure 2). And we provided real data analysis using UK Biobank data (N=30K or 437K) as a training set and either UK Biobank (a separate N=20K sample set) or NHS/HPFS/PHS cohort (N=20K) as a validation set (new Figure 3). Lastly, scatter plots of actual height vs. predicted height were presented (new Figure 4). We believe these new results and the flow of presentation now align the reviewer's suggestions.

Reviewers' comments:

Reviewer #1 (Remarks to the Author):

This latest iteration of the CTPR manuscript wisely pivots to deal with prediction in gigantic cohorts like the UK Biobank where existing prediction methods using individual data (STGBLUP and MTGBLUP) become computationally infeasible. The simulation and real data analyses show notable improvements in prediction accuracy over competing methods that can be applied to Biobank-scale data, such as the summary-statistic methods LDpred and MTAG. I can see the method being quite useful in many large-scale studies. Comments are below.

-For the analysis of UK Biobank and NHS/HPFS/PHS datasets, it would be interesting to include results from other competitors including LD-pred-inf and wMT-SBLUP (Maier et al. : 2018 Nature Communications). Some language on how CTPR differs from wMT-SBLUP would be useful to include in the manuscript.

-For the main simulation results in Figure 2a, it would be useful to include results when the genetic correlation among traits is around 0.15 to mirror the correlations found in the real data examples. Including a variation of Figures 2b and 2c using a correlation of 0.15 would also be useful to include in the supplement.

-For the simulation results in Figure 2, how many causal SNPs were assumed? The language in Page 7 suggests the authors varied the number of causal SNPs between 6.8K and 12.6K but the specific number used for the results in the Figures were not described. If 6.8K causal SNPs were used, could the results for 12.6K SNPs be included in the supplement (or vice versa)?

- A paragraph in the Discussion describing the software package available on the github side would be quite helpful.

Reviewer #2 (Remarks to the Author):

I congratulate the authors for this major revision now utilising the full UK Biobank. I think the manuscript is now much more likely to attract attention and be highly cited.

However, moving to the fullUKB has introduced a new major comment:

The authors use the 457K people from the UK Biobank. This includes close relatives - twins, siblings, parents & offspring, cousins etc. When relatives are excluded the data set is <350K. This is not noted anywhere by the authors and of course has implication for interpretation of the out-of-sample prediction into the UKB, as the 20K retained for the validation sample will have relatives in the training sample. This maybe one reason why the out-of-sample prediction is higher into the UKB than the NHS/HPFS/PHS study. In fact others have used UKB to demonstrate how using information from relatives improves prediction (<https://www.nature.com/articles/srep42091>). It would have been cleaner to use unrelated individuals, but would be a big ask to redo all analyses. At the very least, recognition of inclusion of the relatives is needed in interpreting results. Specifically. It is not clear if this conclusion "Especially, we found that the PA for HGT using Lasso+CTPR (42.8%) captured most of trait-heritability for HGT (45.3%)." is justified. Perhaps this analysis should be redone excluding individuals in the validation sample who are related to those in the training sample.

Minor comments.

- 1) In description of reference 9, I think best to use the nomenclature of the authors wtMT-SBLUP not MTSBLUP
- 2) "In contrast to other methods based on multivariate modelling such as MTGBLUP and MTAG,our method attempts to optimize the prediction accuracy only for the primary trait of interest." This statement is wrong the other methods also generate per individual risks scores for a primary trait (or multiple primary traits simultaneously) using the data from correlated traits. Maybe you are trying to make the same point as the first point on page 13. My interpretation of the wording is different.
- 3) There is a misuse word heritability. The word "trait-heritability" is used for what is commonly called "SNP-heritability". While SNP-heritability may not be ideal, it at least conveys that it is different from the heritability estimated in family studies. Trait-heritability is ambiguous at best and mostly misleading since most would think of trait-heritability as the heritability from family studies.
- 4) Since your UKB data set includes relatives and the GWAS results used in LDSC come from an analysis using SNPTest, which I don't believe acknowledges family structure, then it becomes ambiguous what exactly is estimated in LDSC SNP-heritability.
- 5) Figure S1, C is not defined. Some theory is needed to justified the different maxima for parts a and b. Expected values could be provided also for Fig 2.

Response to Reviewers,

We thank the reviewers for their insightful comments that have allowed us to significantly improve the quality of our manuscript. In this revision, we have re-conducted all the analyses and performed additional analyses to address the comments raised by the reviewers as follows:

First, since our last analyses, a total of 99 participants have requested to withdraw their consent from UK Biobank and thus their data should not be used. The number of overlapping samples with our data is 61 and those samples were removed from our analyses. The number of training samples was changed from 436,898 to 436,837.

Second, in order to more closely reflect the correlation found in the real data analysis, additional simulations with genetic correlation among traits = 0.15 were performed. Overall patterns were similar to the previous simulations and as expected the differences in prediction accuracy between single-trait and multi-trait analyses were smaller than those with larger generic correlation.

Third, to investigate impact of sample relatedness between training and testing sets on potential inflation of prediction accuracy, we re-computed prediction accuracy by removing samples in testing set that are related to any samples in the training set. We compared with previous results and tested whether the difference is significant from 0. With 437K training samples, we observed minimal reduction in prediction accuracy using unrelated testing samples but difference was not statistically significant (change in R^2 from previous to current results ranged from +0.0023 to -0.005, min p-value > 0.422 for H_0 : difference=0).

For your convenience, we provide a point by-point reply to all reviewer's comments with our responses highlighted in blue below.

Reviewer #1 (Remarks to the Author):

This latest iteration of the CTPR manuscript wisely pivots to deal with prediction in gigantic cohorts like the UK Biobank where existing prediction methods using individual data (STGBLUP and MTGBLUP) become computationally infeasible. The simulation and real data analyses show notable improvements in prediction accuracy over competing methods that can be applied to Biobank-scale data, such as the summary-statistic methods LDpred and MTAG. I can see the method being quite useful in many large-scale studies. Comments are below.

We thank the reviewer for the positive comment and finding the usefulness of CTPR and improvement in predictive performance of CTPR over existing methods in large-sample GWAS data.

-For the analysis of UK Biobank and NHS/HPFS/PHS datasets, it would be interesting to include results from other competitors including LD-pred-inf and wMT-SBLUP (Maier et al. : 2018 Nature Communications). Some language on how CTPR differs from wMT-SBLUP would be useful to include in the manuscript.

We appreciate the reviewer's suggestion. As the reviewer suggested, it would be interesting to compare with LDpred-inf and wMT-SBLUP in the manuscript. In our previous results, we have indeed considered LDpred-inf and we clarified further in this revision (see page 8). First, when computing prediction accuracy (PA) using LDpred, we computed PA of LDpred-inf as well as PA of LDpred with a range of ρ values (i.e. tuning parameter for the fraction of causal SNPs, $\rho = 1, 0.5, 0.1, 0.05, 0.01, 0.005, 0.001$). Based on our analyses, PA of LDpred-inf is always worse than LDpred with $\rho < 1$, mostly due to non-infinitesimal architecture of the real and simulation data and hence only PA of LDpred optimized among the ρ values were reported in the manuscript. Second, as demonstrated in the wMT-SBLUP paper (Maier et al. 2018 Nat Comm), PA of wMT-SBLUP is mostly worse than MT-GBLUP because wMT-SBLUP utilizes summary statistics while MT-GBLUP

utilizes individual-level data. Based on our simulation and real data analyses, our CTPR outperformed MT-GBLUP and consequently wMT-SBLUP. Third, LDpred-inf (or SBLUP) and wMT-SBLUP are considered as natural extensions of ST-GBLUP and MT-GBLUP to summary statistics in that both models consider the infinitesimal case (Vilhjalmsson et al. 2015 AJHG, Maier et al. 2018 Nat Comm). Given the above consideration, it is sufficient to show the results from LDpred optimized over the ρ values and the MT-GBLUP as they provided the upper bound of PA for LDpred-inf and wMT-SBLUP, respectively. We now include additional discussion about LDpred-inf and wMT-SBLUP in the result and discussion sections of our manuscript.

-For the main simulation results in Figure 2a, it would be useful to include results when the genetic correlation among traits is around 0.15 to mirror the correlations found in the real data examples. Including a variation of Figures 2b and 2c using a correlation of 0.15 would also be useful to include in the supplement.

As the reviewer requested, we have conducted additional simulations with the genetic correlation among traits=0.15. The result showed similar pattern and conclusion as previous results (left panel below for $\rho=0.15$, $N=30K$, number of traits (T) from 2 to 4, as compared with Figures 2a, 2b; right panel below for $\rho=0.15$, $T=2$, and sample size (N) from 30K to 437K, as compared with Figure 2c). These new results are now included in Figure S11 a and b, as well as Table S11.

-For the simulation results in Figure 2, how many causal SNPs were assumed? The language in Page 7 suggests the authors varied the number of causal SNPs between 6.8K and 12.6K but the specific number used for the results in the Figures were not described. If 6.8K causal SNPs were used, could the results for 12.6K SNPs be included in the supplement (or vice versa)?

We thank for the reviewer's comment. In the manuscript, we did not mention the exact number of causal SNPs for the simulations. Specifically, we used 6.8K causal SNPs for 30K sample analyses, 11.8K for 200K sample analyses and 12.6K for 437K sample analyses to reflect the genetic architecture observed in the real data. We have clarified this further in the main text.

- A paragraph in the Discussion describing the software package available on the github side would be quite helpful.

We added sentences on our CTPR software to the discussion section as “The complete pipeline used to install and execute the CTPR, including required computing resources, input file formats and optional parameters, is provided on GitHub (see the URL below). On top of that, we are implementing the pipeline on public cloud platforms (e.g. Amazon web services, Google cloud platform and Microsoft Azure) such that users do not need to compile and install the pipeline by themselves.”.

Reviewer #2 (Remarks to the Author):

I congratulate the authors for this major revision now utilising the full UK Biobank. I think the manuscript is now much more likely to attract attention and be highly cited.

We thank the reviewer for the positive feedback and the high hope for our method being useful to the broad community.

However, moving to the full UKB has introduced a new major comment:

The authors use the 457K people from the UK Biobank. This includes close relatives - twins, siblings, parents & offspring, cousins etc. When relatives are excluded the data set is <350K. This is not noted anywhere by the authors and of course has implication for interpretation of the out-of-sample prediction into the UKB, as the 20K retained for the validation sample will have relatives in the training sample. This maybe one reason why the out-of-sample prediction is higher into the UKB than the NHS/HPFS/PHS study. In fact others have used UKB to demonstrate how using information from relatives improves prediction (<https://www.nature.com/articles/srep42091>). It would have been cleaner to use unrelated individuals, but would be a big ask to redo all analyses. At the very least, recognition of inclusion of the relatives is needed in interpreting results. Perhaps this analysis should be redone excluding individuals in the validation sample who are related to those in the training sample.

We really appreciate the reviewer’s incisive comment. A total of 6,315 testing samples (31.6%) among 20,000 are inferred to be related (3rd degree or closer) with someone in the full cohort. In order to check how related samples between training and testing sets from UK Biobank have impact on potential inflation of prediction accuracy (PA), we re-computed PA using only unrelated testing samples (N=13,685) and then compared with previous PA using all 20,000 testing samples by testing the difference in PA using Berg et al. 2010, *Statistics & Probability Letters*. The unrelated testing samples (N=13,685) have no relatives (3rd degree or closer) in the cohort and hence there should be no relatives between training and testing sets. With the same 437K training samples, we observed small reduction in PA using unrelated testing samples but difference was not statistically significant based subsampling p-values (change in PA from previous to current results ranged from +0.0023 to -0.005, min p-value >0.422 for H₀: difference=0, see table below for detail). For further comparisons, we randomly selected the same number of testing samples (N=13,685) from NHS/HPFS/PHS cohort and re-computed PA using only these samples. The results were still similar to our previous results and NHS/HPFS/PHS samples still showed smaller PA than UK Biobank testing, likely due to genetic heterogeneity between UK and US samples, and UK Biobank samples were likely environmentally more homogeneous than samples within NHS/HPFS/PHS cohort.

Furthermore, the recent paper (Lello et al. *Genet*, 2018) investigated the impact of relatives in UK Biobank on PA and concluded that there was no discernable difference in prediction results between using a training set drawn from the set of kinship-filtered samples and samples from the unfiltered set. Due to notably large number of related samples (>30%), removing these samples results in a nontrivial decrease in training sample size and PA. Because we found the statistically non-significant impact of related samples in UK Biobank on PA, we retained our results and added explanation on how family structure and cryptic relatedness in UK Biobank affect the PA in the method section and Table S12.

N	Testing	Primary	Second	Method	R ² (All)	R ² (Unrel)	P _{Diff}
30K	NHS/HPFS /PHS	HGT	x	MCP	0.0917	0.0912	0.9820
				LAS	0.1042	0.1038	0.9500
			BMI	MCP+CTPR	0.1132	0.1122	0.8980
				LAS+CTPR	0.1132	0.1123	0.9040
	UKB	HGT	x	MCP	0.1142	0.1151	0.8340
				LAS	0.1313	0.1336	0.6740
			BMI	MCP+CTPR	0.1454	0.1465	0.8760
				LAS+CTPR	0.1458	0.1467	0.8960
437K	NHS/HPFS /PHS	HGT	x	MCP	0.2609	0.2592	0.8540
				LAS	0.2796	0.2779	0.8820
			BMI	MCP+CTPR	0.2922	0.2893	0.7420
				LAS+CTPR	0.2978	0.2944	0.7020
	UKBio	HGT	x	MCP	0.3776	0.3740	0.6040
				LAS	0.3908	0.3862	0.5160
			BMI	MCP+CTPR	0.4248	0.4191	0.4220
				LAS+CTPR	0.4284	0.4234	0.4800

Abbreviations: R² (All): prediction accuracy using all testing samples, R² (Unrel): prediction accuracy using unrelated testing samples, P_{Diff}: subsampling p-values for testing mean difference between R² (All) and R² (Unrel).

Specifically. It is not clear if this conclusion “Especially, we found that the PA for HGT using Lasso+CTPR (42.8%) captured most of trait-heritability for HGT (45.3%).” is justified.

We really thank for the reviewer’s sharp comment. The narrow sense heritability is the theoretical limit of polygenic prediction based on linear model in large sample size assuming that the estimated heritability is a true reflection of the population parameter (Wray et al. 2013, Nat Rev Genet). Because our SNP-heritability for HGT was estimated using LDSC with summary statistics and our PA was computed using CTPR with individual-level GWAS data, estimated SNP-heritability (45.3%) may not necessarily be the theoretical upper limit of our PA (42.8%). In order to make our conclusion much clearer, we revised it as: “Especially, we found that the PA for HGT using Lasso+CTPR (42.8%) captured most of estimated SNP-heritability for HGT (45.3%) using the same UK Biobank data although our SNP-heritability estimate may be underestimated (because current SNP-heritability estimate ranges from 45% [Yang et al. 2010, Nat Genet] to 54% [Rawlik et al 2016, Genome Biol]) and thus 45.3% may not be the upper limit of the PA”.

Minor comments.

1) In description of reference 9, I think best to use the nomenclature of the authors wtMT-SBLUP not MTSBLUP

Thanks for the clarification on the nomenclature of wMT-SBLUP (approximate multi-trait summary statistic BLUP using a weighted index). We replaced MTSBLUP by wMT-SBLUP as requested.

2) “In contrast to other methods based on multivariate modelling such as MTGBLUP and MTAG, our method attempts to optimize the prediction accuracy only for the primary trait of interest.” This statement is wrong the other methods also generate per individual risks scores for a primary trait (or multiple primary traits simultaneously) using the data from correlated traits. Maybe you are trying to make the same point as the first point on page 13. My interpretation of the wording is different.

As the reviewer mentioned, other multivariate models such as MTGBLUP and MTAG generate individual risk scores for a primary trait or multiple traits as well. However, our CTPR focuses more on improvement of the PA for a primary trait of interest and thus, to select the best tuning parameter using cross-validation, CTPR only

divide the dataset for the primary trait into n folds and each fold is used as the validation set and the remaining as the training set. The PA for secondary traits are not computed and will not contribute to the fitting procedure of CTPR. Only when a part of samples has data on both the primary and secondary traits, the secondary ones are also divided into n folds to avoid overfitting. We described this in details in the method section (see page 23).

3) There is a misuse word heritability. The word “trait-heritability” is used for what is commonly called “SNP-heritability”. While SNP-heritability may not be ideal, it at least conveys that it is different from the heritability estimated in family studies. Trait-heritability is ambiguous at best and mostly misleading since most would think of trait-heritability as the heritability from family studies.

We thank the reviewer’s sharp comment. As requested, we replaced ‘trait-heritability’ by ‘SNP-heritability’ in the manuscript.

4) Since your UKB data set includes relatives and the GWAS results used in LDSC come from an analysis using SNPTest, which I don’t believe acknowledges family structure, then it becomes ambiguous what exactly is estimated in LDSC SNP-heritability.

We showed that 45.3% (s.e.=4.1%), 24.3% (s.e.=2.4%), 20.1% (s.e.=2.1%) and 18.7% (s.e.=1.9%) of variance of HGT, BMI, HIP and WST, respectively, can be explained by 1M SNP data from the UK Biobank. We estimated these SNP-heritability using LDSC with summary statistics generated by SNPTest. For summary statistics, we first removed samples who have more than 10 putative third-degree relatives and then fitted linear model with top 10 genotype PCs. We understand this cannot fully account for family structure of UK Biobank samples but it helps to remove most genetic relatedness among them. For comparison, from Neale lab’s GWAS results based on 337K unrelated individuals (<http://www.nealelab.is/uk-biobank/>), heritability estimate for HGT, BMI, HIP and WST are 46.2%, 24.6%, 22.3% and 20.4%. We found that these estimates are similar to ours although these are slightly larger than ours because they were calculated using only samples with British ancestry, not all European ancestry.

5) Figure S1, C is not defined. Some theory is needed to justified the different maxima for parts a and b. Expected values could be provided also for Fig 2.

In Figure S1, C is the number of causal SNPs. Based on the theory in Yang et al. 2010 Nat Genet, the theoretical upper bound of PA is given by $h^2/[1+C/h^2N]$ where h^2 : heritability, C : number of causal SNPs and N : samples size. The difference in upper bound of PA between Figure S1a and Figure S1b resulted from the difference in the number of causal SNPs ($C=500$ vs 1000). Also, two upper bounds for each plot represent upper bounds for single-trait (lower) and multi-trait approaches (upper). Because half of training samples were used for the primary trait and the other half for the secondary trait, multi-trait approaches utilize double sample size than single-trait ones assuming a genetic correlation of 1.

The theoretical upper bounds of PA for Figure 2 are given by 0.299 for simulations with 30K samples ($C=6,800$, $h^2=0.45$), 0.398 with 200K samples ($C=11,800$, $h^2=0.45$) and 0.423 with 437K samples ($C=12,600$, $h^2=0.45$).

REVIEWERS' COMMENTS:

Although there are no Remarks to the Author from Reviewer #1, in his/her Remarks to the Editor, this reviewer says that he/she is satisfied with the manuscript.

Reviewer #2 (Remarks to the Author):

Thank you for the detailed responses.

I realise it was a lot of work to redo analyses excluding relatives, and it is useful to see that the relatives had little impact.

In point 5) C =causal number of SNPs assuming the causal SNPs are known and only this number of SNPs are tested. If the causal SNPs are not known a priori this number is the effective number of SNPs tested, so ~60K for genome-wide SNP data. So I guess you have provide theoretical maximum, but could be misleading without qualification

Response to Reviewers,

We thank the reviewers for their positive remarks on our manuscript. This revision addressed the final reviewers' comments as well as all the editorial requests we have received. For your convenience, we have made all our minor textual updates on the draft using track review changes in Microsoft Word.

Reviewer #1 (Remarks to the Author):

Although there are no Remarks to the Author from Reviewer #1, in his/her Remarks to the Editor, this reviewer says that he/she is satisfied with the manuscript.

We are really grateful for the reviewer's satisfaction with the revised manuscript.

Reviewer #2 (Remarks to the Author):

Thank you for the detailed responses.

I realise it was a lot of work to redo analyses excluding relatives, and it is useful to see that the relatives had little impact.

In point 5) C =causal number of SNPs assuming the causal SNPs are known and only this number of SNPs are tested. If the causal SNPs are not known a priori this number is the effective number of SNPs tested, so ~60K for genome-wide SNP data. So I guess you have provide theoretical maximum, but could be misleading without qualification

We appreciate the reviewer's comment and fully agree with the remark. As indicated, when the number of causal SNPs (C) is known a priori and other SNPs are completely irrelevant to causal ones and not tested, the theoretical maximum becomes the real maximum. For simulations, we used 5,000 SNPs that are in low LD from the NHS/HPFS/PHS cohort and assumed only causal SNPs (500 or 1,000) have effects from the multivariate normal distribution (i.e. $\beta_j \sim N(\mathbf{0}, \mathbf{D})$) while non-causal ones have no effects (i.e. $\beta_j = \mathbf{0}$). Because all the methods we compared use not just the causal SNPs but all 5,000 SNPs, the real maximum might be smaller than the theoretical maximum. We added the following sentences for clarification to the supplementary document (page 1): "Note: this theoretical upper bound of prediction accuracy is achievable if and only if the causal SNPs are used in the prediction model. When causal SNPs are not known a priori, the theoretical upper bound could be lower."